# Broad Whitefish (*Coregonus nasus*) isotopic niches: Stable isotopes reveal diverse foraging strategies and habitat use in Arctic Alaska

**Jason C. Leppi** [1,2]*, **Daniel J. Rinella**[3], **Mark S. Wipfli**[4], **Matthew S. Whitman**[5]

**1** Alaska Cooperative Fish and Wildlife Research Unit, College of Fisheries and Ocean Sciences, University of Alaska Fairbanks, Fairbanks, Alaska, United States of America, **2** Research Department, The Wilderness Society, Anchorage, Alaska, United States of America, **3** Anchorage Fish and Wildlife Conservation Office, U. S. Fish and Wildlife Service, Anchorage, Alaska, United States of America, **4** U.S. Geological Survey, Alaska Cooperative Fish and Wildlife Research Unit, Institute of Arctic Biology, University of Alaska Fairbanks, Fairbanks, Alaska, United States of America, **5** Arctic District Office, Bureau of Land Management, Fairbanks, Alaska, United States of America

* jcleppi@alaska.edu, jason_leppi@tws.org

**Data Availability Statement:** All relevant data are within the paper and its Supporting information files.

## Abstract

Understanding the ecological niche of some fishes is complicated by their frequent use of a broad range of food resources and habitats across space and time. Little is known about Broad Whitefish (*Coregonus nasus*) ecological niches in Arctic landscapes even though they are an important subsistence species for Alaska's Indigenous communities. We investigated the foraging ecology and habitat use of Broad Whitefish via stable isotope analyses of muscle and liver tissue and otoliths from mature fish migrating in the Colville River within Arctic Alaska. The range of $\delta^{13}C$ (-31.8– -21.9‰) and $\delta^{15}N$ (6.6–13.1‰) across tissue types and among individuals overlapped with isotope values previously observed in Arctic lakes and rivers, estuaries, and nearshore marine habitat. The large range of $\delta^{18}O$ (4.5–10.9‰) and $\delta D$ (-237.6– -158.9‰) suggests fish utilized a broad spectrum of habitats across elevational and latitudinal gradients. Cluster analysis of muscle $\delta^{13}C'$, $\delta^{15}N$, $\delta^{18}O$, and $\delta D$ indicated that Broad Whitefish occupied four different foraging niches that relied on marine and land-based (i.e., freshwater and terrestrial) food sources to varying degrees. Most individuals had isotopic signatures representative of coastal freshwater habitat (Group 3; 25%) or coastal lagoon and delta habitat (Group 1; 57%), while individuals that mainly utilized inland freshwater (Group 4; 4%) and nearshore marine habitats (Group 2; 14%) represented smaller proportions. Otolith microchemistry confirmed that individuals with more enriched muscle tissue $\delta^{13}C'$, $\delta D$, and $\delta^{18}O$ tended to use marine habitats, while individuals that mainly used freshwater habitats had values that were less enriched. The isotopic niches identified here represent important foraging habitats utilized by Broad Whitefish. To preserve access to these diverse habitats it will be important to limit barriers along nearshore areas and reduce impacts like roads and climate change on natural flow regimes. Maintaining these diverse connected habitats will facilitate long-term population stability, buffering populations from future environmental and anthropogenic perturbations.

**Funding:** This research was funded in-part by the National Science Foundation [Awards Alaska EPSCoR OIA-1208927; ARCSS-1722572] and the state of Alaska in the form of funds to MSWi and by the U.S. Bureau of Land Management [Interagency Agreement Number L15PG00216] in the forms of funds to MSWh. The Wilderness Society provided support in the form of a salary for JCL, the U.S. Geological Survey in the form of a salary to MSWi, the U.S. Bureau of Land Management in the form of a salary to MSWh, and the U.S. Fish and Wildlife Service in the form of a salary to DJR. The specific roles of these authors are articulated in the 'author contributions' section. In-kind support such as borrowed equipment and staff assistance was provided by the U.S. Fish and Wildlife Service and the Native Village of Nuiqsut Tribal Council. The funders had no role in the study design, data collection, and analysis, decision to publish, or preparation of the manuscript.

**Competing interests:** The authors have declared that no competing interests exist.

# Introduction

The ecological niche conceptualizes the physical environment and food resources used by a species [1]. First described by Grinnell (1917) [2], the "niche habitat" concept proposed that environmental conditions across geographic space limit, or at least influence, the habitat utilized by species. Elton's "niche" concept further explored this idea but focused on resources consumed and the status of a species within a community, including trophic relationships between prey and predators [3]. Building from previous definitions, Hutchinson (1957) [4] described the "fundamental niche" as a set of points in multivariate space whose axes represent both physical and biological variables required by a species. Within the fundamental niche concept, the first axis represents the bioclimate variables or the habitat stage used by a species and the second axis represents the prey resources that the species consumes [4]. Inside the fundamental niche is the "realized niche," which is constrained by predation and competition.

Diverse foraging behaviors and anadromy help fishes maximize foraging efficiency by enabling them to exploit a suite of seasonally available habitats and food supplies [5, 6]. The evolutionary basis for anadromy in high-latitude fishes is linked to generally lower freshwater productivity compared to marine environments [7, 8]. In addition, extreme seasonal variation in climate and changing hydrologic conditions create a shifting and heterogeneous mosaic of food-resources and suitable habitats [9] that can be exploited by mobile generalists typical of Arctic and boreal fishes [10–13]. The benefits of diverse foraging behavior within Arctic fishes are also influenced by ontogenetic changes in diet that favor migration between habitats to maximize prey intake and minimize energetic costs [14–16]. Collectively, these factors complicate the task of characterizing the ecological niches of Arctic fishes.

Many animals inhabit ecosystems that are difficult to monitor and frequently move in search of food (e.g., fishes), which makes it difficult to quantify niche space via conventional techniques. Such techniques require extensive sampling to accurately measure diet composition yet generally lack temporal integration and fail to account for variation in assimilation rates [17, 18]. Alternatively, stable isotope analysis offers an approach for characterizing ecological niches that can time-integrate multiple dimensions of information on both resources and habitat [19–21]. Stable isotopes change in systematic ways within and across ecosystems [22, 23] and are incorporated into animals' tissue through food and environmental water. Stable isotope ecologists can gain new insights into what a species consumes and where it lives through the development of the "isotopic niche" [19], which uses multiple stable isotope ratios within tissues to characterize a species' niche space and the location of individuals within that space [17, 19, 24, 25]. Stable isotope analyses of carbon ($\delta^{13}C$) and nitrogen ($\delta^{15}N$) from tissues with different turnover rates (e.g., muscle, liver) can be used to understand diet and trophic position over different time spans [26–28]. Oxygen ($\delta^{18}O$) and deuterium ($\delta D$) isotope ratios in animal tissues change in predictable ways across landscapes and, when used in combination with modeled isoscapes [29], can offer additional insights into habitat use [18, 30].

Otolith microchemistry is another tool to help understand ecological niches of highly mobile fish species [31–37]. Otoliths, paired inner ear stones used for hearing and balance in all teleost fishes, are laid as concentric layers of metabolically inert biogenic minerals, primarily calcium carbonate. Elements are permanently incorporated into their organic matrix, and compositional changes across the layers reflect changes across an individual's life [38]. Strontium (Sr), a naturally occurring element derived from geologic material, has four stable isotopes ($^{88}Sr$, $^{87}Sr$, $^{86}Sr$, $^{84}Sr$), in which only $^{87}Sr$ is radiogenic. The ratio of $^{87}Sr$ to $^{86}Sr$ ($^{87}Sr/^{86}Sr$) reflects Sr released into fresh water sources and is driven by differences in lithology, age, chemical composition [39–41], and weathering rates of surficial geology [42–44]. For diadromous fishes, the relative differences in $^{87}Sr/^{86}Sr$ between freshwater and isotopically uniform marine

values help provide detailed information on the timing and duration of estuarine and marine habitat use [45, 46].

Arctic Alaska is undergoing major landscape and ecosystem transformations from climate change [47–51] and oil and gas development [52–54]. Arctic surface air temperatures are warming at more than twice the rate of lower latitudes, which is exacerbated by Arctic amplification—the feedback between air temperature and surface albedo in polar regions [55]. The accelerated impacts of climate change at high latitudes [56, 57] are a major threat to Arctic freshwater ecosystems [58], altering streamflow patterns [59–62], warming [63, 64] and drying [65] aquatic habitats, causing eutrophication [66] and browning of lakes [67, 68], and allowing for northward range expansion of eurythermic species [69]. Warmer air and fewer cold days have led to numerous changes in the Arctic cryosphere [70], including degraded permafrost [71] and increased active layer depth [72], ground subsidence and alterations in the patterned ground features [54], and increasing retrogressive thaw slump activity [73, 74]. These changes in permafrost and seasonally frozen ground have resulted in increased riverine nutrient [75] and sediment loads [76, 77] in freshwater ecosystems.

Broad Whitefish (*Coregonus nasus*) is a primary subsistence resource for Indigenous peoples in Alaska. Referred to as *Aanaakliq* in the Iñupiaq language, Broad Whitefish are valued due to their relatively large size (up to 4.5 kg) and abundance during migrations, and account for about half the total mass of fishes harvested across all Beaufort Sea communities [78, 79]. However, information on Broad Whitefish habitat use in Arctic landscapes is limited. Previous research supports the theory of a highly mobile species that utilizes a variety of aquatic habitats [80, 81]. To rapidly build energy reserves during the brief open water period, it is likely that Broad Whitefish move across the landscape and use a variety of habitats (e.g., lakes, rivers, streams, estuaries, lagoons, and nearshore marine areas) [80, 81] while feeding on a diversity of benthic and pelagic prey across a range of trophic levels [12]. Connectivity between a variety of habitats is, therefore, especially important for Broad Whitefish. Similar to other Arctic fishes, habitat use across time and space results in a variety of life histories [81] with varying amounts of time spent in freshwater, estuarine and marine habitats [80] to maximize growth, survival, and reproduction. As such, Broad Whitefish can be considered a model species, which can help us understand habitat use of other similar Arctic fish species (e.g., Humpback Whitefish *Coregonus pidschian*).

Broad Whitefish populations use the Colville River watershed for foraging, rearing [82, 83] and spawning [82]. With headwaters in the rugged Brooks Range, the Colville River is one of the few rivers in the region that contains abundant gravel substrate and deep channels, which are both likely essential for egg survival [84]. Due in part to its watershed size, the Colville River also has the largest delta on the Alaskan Beaufort Sea coast, which provides abundant rearing habitat for larval and juvenile fishes [80, 83]. Broad Whitefish can live for 30+ years and return to the Colville River ecosystem regularly to reproduce [85], likely migrating from a variety of productive foraging areas in rivers across the Beaufort Coastal Plain. Thus, by sampling the Colville River's spawning run, we were able to infer patterns of foraging behavior and habitat use for Broad Whitefish at the regional scale.

Food resources are dispersed across space and time within high-latitude aquatic ecosystems, which should favor fishes with generalist foraging strategies and the ability to exploit a variety of habitat types [11, 12]. Extreme seasonal variation in climate and changing hydrologic conditions, across a spectrum of freshwater to marine habitats, likely creates a variety of seasonally productive habitats and a diversity of ecological niches for Broad Whitefish and other Arctic fishes. We investigated the ecological niches utilized by Broad Whitefish—a highly mobile, generalist fish species—in the Alaska Arctic. Our specific objectives were to (1) explore how variation in $\delta^{13}C$ and $\delta^{15}N$ in Broad Whitefish muscle relates to Arctic freshwater and marine

foraging niches (2) investigate whether diet changes across the summer by comparing tissues with different isotopic turnover rates, (3) determine how variation in muscle $\delta^{18}O$ and $\delta D$ relates to Arctic freshwater and marine niches (4) characterize isotopic niches using cluster analysis of stable isotope ratios ($\delta^{13}C$, $\delta^{15}N$, $\delta^{18}O$, and $\delta D$) from muscle tissue and investigate the relationship between the resulting isotopic cluster groups and potential niches utilized, and (5) determine if stable isotope values from muscle tissue are predictive of life history strategy by comparing to Sr isotopes in otoliths. Our research revealed new insights into Broad Whitefish isotopic niches and provided new information on important habitat and food resource use by Broad Whitefish within the Beaufort Sea region. This information expands our understanding of the mosaic of feeding habitats used and will better inform management and conservation decisions to protect this vital subsistence resource.

## Materials and methods

### Study area

The central Beaufort Sea region study area (Fig 1) contains a diversity of foraging habitats for Arctic fishes. Situated between the Ikpikpuk and Canning rivers, the coastline is a spectrum of deep bays and inlets, tapped lake basins (lake basins that are breached by the sea due to erosion), lagoons with barrier islands, and exposed bluffs [86]. River deltas of varying size are frequent along the coast [87]. Thermokarst and riverine lakes that vary in size, depth, and connectivity [87, 88] account for 30% of the region's surface area [89]. Stream habitats vary by watershed and geomorphic setting [90, 91], generally resulting in colluvial channels in foothill and mountainous headwaters, beaded headwater streams in low-gradient coastal plains, and meandering alluvial streams and rivers lower in watersheds [88].

The region, within the Arctic tundra biome, is characterized by permafrost, extreme climate, low-growing plants, and large variations in day length. The region's stark seasonality can be divided into a long cold season and a short warm season, but the former controls many of the physical and biological processes. Cold season air temperatures are consistently well below freezing, creating a landscape dominated by snow and ice for about eight months [92]. The warm season is brief, but with 24 hours of daylight and moderate air temperatures [92], the area becomes productive foraging and rearing habitat for many resident and migratory fishes, mammals, and birds. Annual precipitation is generally low, with more falling in the foothills than along the coast (30 and 20 cm, respectively) and about half falling as snow [92].

### Fish sampling

We collected otoliths and tissue from adult Broad Whitefish migrating up the Colville River during 2015. The Colville, the largest river in Arctic Alaska, flows about 560 km northward from its headwaters in the partially glaciated Brooks Range to a large delta on the edge of the central Beaufort Sea coast, near the Alaska Native village of Nuiqsut (Fig 1). We set gill nets ca. 30 m in length, composed of braided nylon and monofilament with 10-cm and 12-cm stretched mesh, to target adult fish large enough to spawn (> 35 cm) [93]. We positioned nets at gravel point bars, along eddy lines, and perpendicular to flow in low-gradient reaches at three separate sites (Sites 1–3; Fig 1). We sampled at Puviksuk on July 23–27, at Umiat on August 21–26, and at Itkillik on October 10–11. We euthanized captured Broad Whitefish with a single sharp blow to the cranium and recorded fork length (n = 98, all of which were adults ≥ 42 cm; [84], total weight, gonad weight, and sex (44 males, 47 females, 7 undetermined; S1 Table). We also collected liver and epaxial muscle samples with sterile 5-mm biopsy punches (preserved with clay desiccant beads) from all individuals, except in cases where fish organs were consumed by birds while caught in gillnets (n = 5). Liver and muscle tissue likely

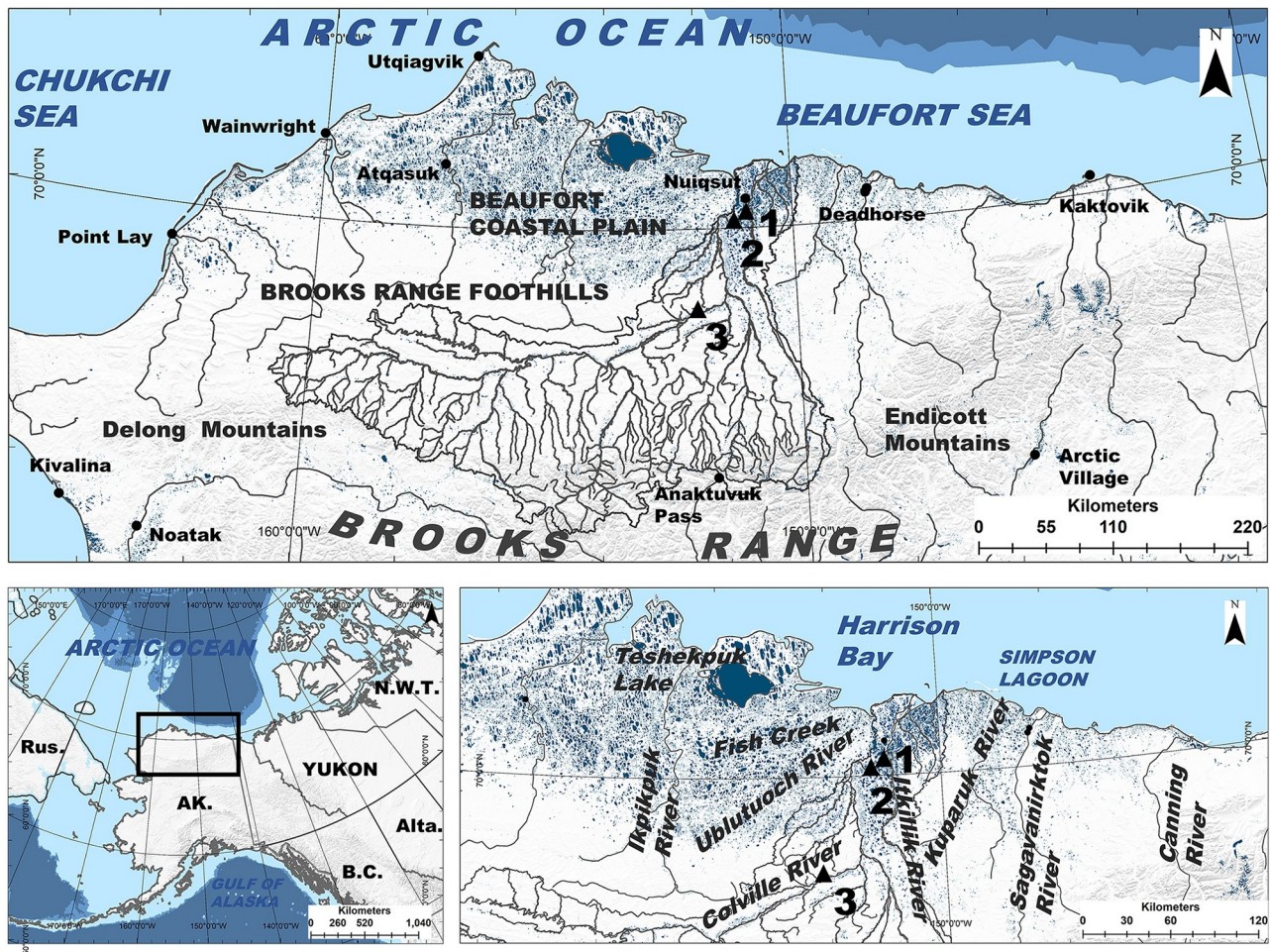

**Fig 1. Study area.** The central Beaufort Sea region in Arctic Alaska, situated between the Ikpikpuk River and the Canning River, contains a diversity of aquatic habitat. The large Colville River, AK, USA (ca. watershed area 60,000 km²), located in the middle of the central Beaufort Sea coast, contains minor tributaries that drain from the Brooks Range (thin grey lines) and main tributaries (thick dark grey lines) that flow toward a large delta on the edge of the Beaufort Sea, near the community of Nuiqsut, AK. Fish were collected at three sites (black triangles) within the Colville River (site 1 = Itkillik, site 2 = Puviksuk, site 3 = Umiat). Data sources: USGS National Map Viewer (http://viewer.nationalmap.gov/viewer/), Natural Earth (http://www.naturalearthdata.com/), National Hydrography Dataset (https://www.usgs.gov/national-hydrography/national-hydrography-dataset).

represent the integration of consumed food resources for ca. 37 and 88 days prior to capture [26], respectively, providing records of fish foraging niches for differing periods during the growing season. Adult Broad Whitefish are slow-growing fish and it is likely that integration of isotopes into tissues is primarily through anabolism, which could mean that integration of food resources is much longer [27]. Sagittal otoliths were collected from each individual using the Guillotine method [94], rinsed in water, and stored in paper envelopes. The planned sample size of 50 individuals per site was lower than anticipated at Umiat (n = 23) and Itkillik (n = 17), as opposed to Puviksuk (n = 57), due to unexpectedly high streamflow at the former and an early freeze-up at the latter that inhibited our ability to capture fish. Research was conducted under Bureau of Land Management NPR-A permit #FF097006 and the Alaska Department of Fish and Game, Fish Resource permit #SF2015-200. All collections were performed using methods in line with guidelines to minimize suffering.

## Tissue stable isotope analyses and data analysis

We selected stable isotopes $\delta^{13}C$, $\delta^{15}N$, $\delta^{18}O$, and $\delta D$ due to their combined ability to discern Broad Whitefish diet, trophic position, and habitat use [22–25, 30, 95, 96]. We analyzed tissues with different turnover rates (e.g., muscle, liver) to understand change in stable isotopes over different time spans [26–28]. Both $\delta^{13}C$ and $\delta^{15}N$ have remained important stable isotopes for reconstructing foraging ecology patterns due to their exclusive association with diet, limited fractionation, and ability to reflect sources of primary production and trophic position [22, 97].

While relatively new to food web studies, tissue $\delta^{18}O$ and $\delta D$, when used in combination with modeled isoscapes, can offer additional insights into diet and habitat use [18, 30, 95, 97].

We analyzed liver and muscle tissue at the University of Alaska Anchorage's Environment and Natural Resources Institute (ENRI) Stable Isotope Facility. Samples were dried, ground to a fine powder, and weighed to 0.001 g prior to analysis. Liver and muscle samples were analyzed for $\delta^{13}C$ and $\delta^{15}N$ using a Costech ECS 4010 elemental analyzer (Costech, Valencia CA) in line with a Thermo Finnigan™ Delta V continuous-flow isotope ratio mass spectrometer (Thermo Scientific™, Bremen, Germany). Muscle samples were analyzed for $\delta D$ and $\delta^{18}O$ using a Thermo Finnigan TC/EA in line with a Thermo Finnigan™ Delta Plus™ XP continuous-flow isotope ratio mass spectrometer (Thermo Scientific™, Bremen, Germany). Due to the reduced size of dried liver samples, we were unable to analyze liver tissue for $\delta D$ and $\delta^{18}O$. Instruments were calibrated against international reference standards from the International Atomic Energy Agency and the United States Geological Survey. Stable isotope compositions were referenced relative to international standards; atmospheric N for nitrogen, Vienna Pee Dee Belemnite (VPDB) for carbon and Vienna standard mean ocean water (VSMOW) for oxygen and deuterium. Stable isotope ratios were expressed in $\delta$ notation in units of per mil (‰) relative to international standards where: $\delta X = [(R_{sample}—R_{standard})/R_{standard}]^* 1000‰$, where R is the ratio between the isotopes (i.e., $^{13}C/^{12}C$, $^{15}N/^{14}N$, $^{18}O/^{17}O$, $^2H/^1H$). Long-term records of internal standards yield an analytical precision of 0.11 ‰ for $\delta^{15}N$, 0.12 ‰ for $\delta^{13}C$, 0.2 ‰ for $\delta^{18}O$, and 1.8 ‰ for $\delta D$.

To account for a subset of values with carbon to nitrogen ratios (C:N) greater than 3.5, we lipid-normalized $\delta^{13}C$ samples following an approach outlined in Skinner et al. (2016) [98]. Using their approach, we adjusted $\delta^{13}C$ using the Kiljunen et al. (2006) [99] mathematical normalization model ($\delta^{13}C'$(normalized $\delta^{13}C$) = $\delta^{13}C$ + D (I + (3.90)/(1 + 278/L)) with difference in carbon isotopic composition between protein and lipid (D) equal to 7.018 and the constant (I) equal to 0.048. We calculated percent lipid (L) using the Post et al. (2007) [100] equation (L = -20.54 + 7.34 × C:N). We statistically analyzed the stable isotope ratios from Broad Whitefish tissue samples through a hierarchal clustering approach to characterize ecological niches [101]. Hierarchal clustering builds a hierarchy of clusters more similar to each other and, in this instance, our approach clustered individuals into groups with similar isotopic ratios, which represented an individual's habitat and food resources in multivariate space. Assigning clusters is complicated due to the high-dimensional nature of biological data, which makes it difficult to visualize, but overall, clustering can help gain insights [101]. We conducted a hierarchal agglomerative clustering analysis on normalized-rescaled values following methods outlined by Charrad et al. (2014) [102] within R statistical program using the NbClust package, average link method, and Euclidian distance. Next, we generated 30 cluster validity indices available within the NbClust package to assess the optimal number of clusters (groups) and used the majority rule to determine the best number of clusters [102].

To determine if an individual's diet remained stable or changed over the summer period, we visually and quantitatively compared the difference between muscle and liver tissues (i.e., muscle minus liver) for both $\delta^{15}N$ and $\delta^{13}C$. We also conducted a statistical correlation analysis in

R statistical program using the ggpubr package. To assess the data for normality we used quantile-quantile plots and the Shapiro-Wilk's test. Then, we conducted a Pearson correlation analysis between muscle and liver $\delta^{15}N$ and $\delta^{13}C$ to determine the linear correlation between the muscle and liver tissues. Last, we conducted a linear regression analysis for the muscle and liver $\delta^{15}N$ and $\delta^{13}C$ to determine the slope and statistical significance of the best-fit line.

## Otolith microchemistry and life history classification

We measured Sr isotope concentrations ($^{88}Sr$, $^{87}Sr$, $^{86}Sr$, $^{84}Sr$) across a subset of the otoliths (69 of 98 individuals sampled). In preparation for isotope analysis, we mounted otoliths prepared in the transverse plane on petrographic slides following methods outlines in Leppi 2021 [82]. We used an Analyte G2 Excimer 193-nm Laser Ablation System (LA; Teledyne Photon Machines, Bozeman, USA) with a Helex cell coupled to a Neptune Plus™ multi-collector inductively coupled plasma mass spectrometer (MC-ICP-MS; Thermo Scientific™, Bremen, Germany) for strontium isotope analyses at the University of Alaska Fairbanks, Alaska Stable Isotope Facility following methods outlined in Leppi 2021 [82]. Briefly, we selected samples to prioritize for laser ablation with the goal of analyzing otoliths across a gradient of $\delta^{13}C'$, $\delta D$, and $\delta^{18}O$, which was expected to represent time spent in different habitat types (e.g., freshwater, estuarine, marine) over the three months prior to capture [82]. Subsampled otoliths were roughly proportional to the number of samples collected at each field site [82]. Compared to freshwater habitats, $^{87}Sr/^{86}Sr$ in marine habitats is generally lower, homogenous, and constant due to the long residence time and mixing of oceans [45, 46].

We calculated Sr concentrations (Sr mg/kg) by dividing the concentration of Sr in the FEBS-1 standard (i.e., 2055 mg/kg) by the average $^{88}Ca$ FEBS-1 standard value during ablation and multiplying by otolith $^{88}Ca$ at individual points across the ablation path. We considered otolith $^{88}Sr$ below 6.13 voltage V (ca. 850 mg/kg) to be time spent in freshwater, greater than 12.26 V (ca.1700 mg/kg) to be marine habitat use [45, 46], and intermediate values to reflect time spent in estuarine habitat. The $^{88}Sr$ concentrations systematically increase with water salinity, with lower values found in freshwater compared to marine habitats. For diadromous fishes, these relative differences in Sr isotopes between distinctive freshwater, estuarine, and marine values can be used to indicate the timing and duration of habitat use [45, 46]. If Sr data were highly variable or contained unreliable values due to cracks, otoliths were removed from the dataset (n = 6).

We visually compared the $^{88}Sr$ and $^{87}Sr/^{86}Sr$ across each otolith core-to-edge chronology and used a supervised classification approach to group otoliths into three life history groups. Life history types included anadromous, semi-anadromous, and nonanadromous. We considered individuals with maximum $^{88}Sr$ above 12.26 V and $^{87}Sr/^{86}Sr$ near the global mean oceanic value (GMV = 0.70918 ± 0.00006 2 standard deviations (SD)) anadromous. We classified individuals as semi-anadromous if $^{87}Sr/^{86}Sr$ at the natal region was near GMV. Semi-anadromous individuals had no detectable age-0 freshwater otolith signature, likely spending limited time in freshwater as larvae and frequently moving between freshwater, estuarine, and marine habitats [82]. Nonanadromous individuals had $^{88}Sr$ concentrations lower than 12.26 V and $^{87}Sr/^{86}Sr$ consistently below global marine $^{87}Sr/^{86}Sr$, indicating that they did not enter marine habitats.

# Results

## $\delta^{13}C'$ and $\delta^{15}N$ in isotope space

Stable isotope ratios measured in Broad Whitefish muscle tissue overlapped with those previously observed from a variety of Arctic plants, invertebrates, and fish species (Fig 2). Previous

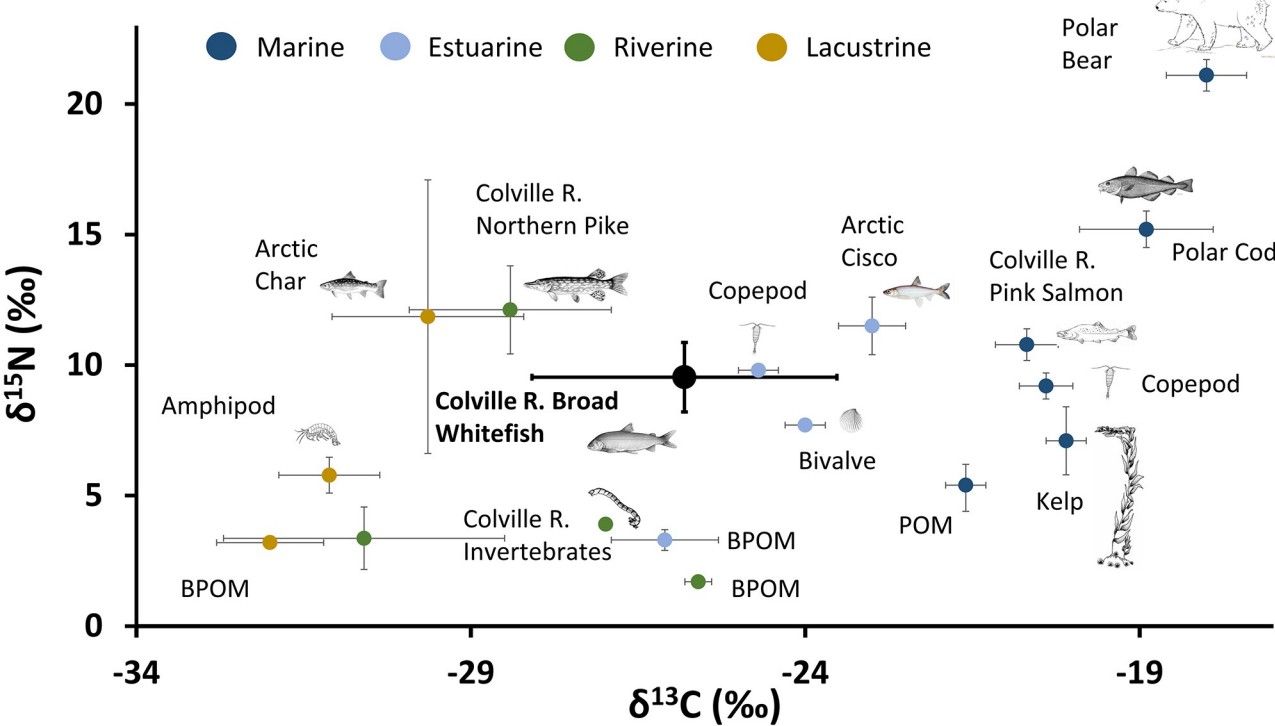

**Fig 2. Arctic isoscape.** Cross-section of Arctic aquatic ecosystems. Circles with error bars represent the mean and standard deviation of the organism from each type of ecosystem (Dark blue = marine, light blue = estuarine, green = riverine, gold = lacustrine). Error bars on estuarine species and benthic particulate organic matter values represent the standard error. Circles without error bars represent a single sample. For comparison, data from outside the study area is shown: marine samples are from Admiralty Inlet in the Northwest Territories, CA; lagoon samples are from sites along the eastern Beaufort Sea coast, Alaska, USA; riverine samples are from the lower MacKenzie River, Yukon, CA, and Lacustrine samples are from Toolik Lake, Alaska, USA. Additional isotope sources: See S2, S3 Tables.

research has shown that species that inhabited inland lakes and rivers had more depleted $\delta^{13}C$, compared to marine species (See S2 Table). Broad Whitefish normalized muscle values ($\delta^{13}C'$) ranged from -31.8 to -21.9‰ and overlapped with $\delta^{13}C$ observed in Arctic lakes and rivers, estuaries, and nearshore marine habitats (Fig 2) [103–105]. The mean $\delta^{13}C'$ of -25.8‰ was roughly between values observed in several riverine and estuarine species (Fig 2; S2, S3 Tables). Muscle $\delta^{15}N$ was less variable than observed $\delta^{13}C'$ ($\delta^{15}N$ range = 6–13.1‰; Fig 4) and was generally higher than that of invertebrates and lower than other fishes (Fig 2) [103–105].

## Tissue comparison

The difference between muscle and liver tissues (i.e., muscle minus liver) was generally minor, with most individuals having a disparity of < 2.0‰ for both $\delta^{13}C'$ and $\delta^{15}N$. Differences for $\delta^{13}C'$ ranged from -6.75 to 2.29‰ (mean -0.13‰; S.D. = 1.24‰; Fig 3) while differences for $\delta^{15}N$ ranged from -1.50 to 3.97‰ (mean 0.29‰; S.D. = 0.81‰; Fig 3). Delta $\delta^{13}C'$ and $\delta^{15}N$ were significantly correlated (p-value < 0.05, R = 0.85 and 0.82, respectively) and indicated a moderately strong positive relationship between muscle and liver isotope ratios ($\delta^{13}C'$ = p-value < 0.001, $\beta$ = 0.9, $R^2$ = 0.73; $\delta^{15}N$ = p-value < 0.001, $\beta$ = 0.85, $R^2$ = 0.66). However, the disparity between muscle and liver isotope ratios suggests that the diet of two individuals shifted toward prey items more depleted in $\delta^{13}C$, while that of two others shifted toward items more enriched in $\delta^{15}N$ (Fig 3).

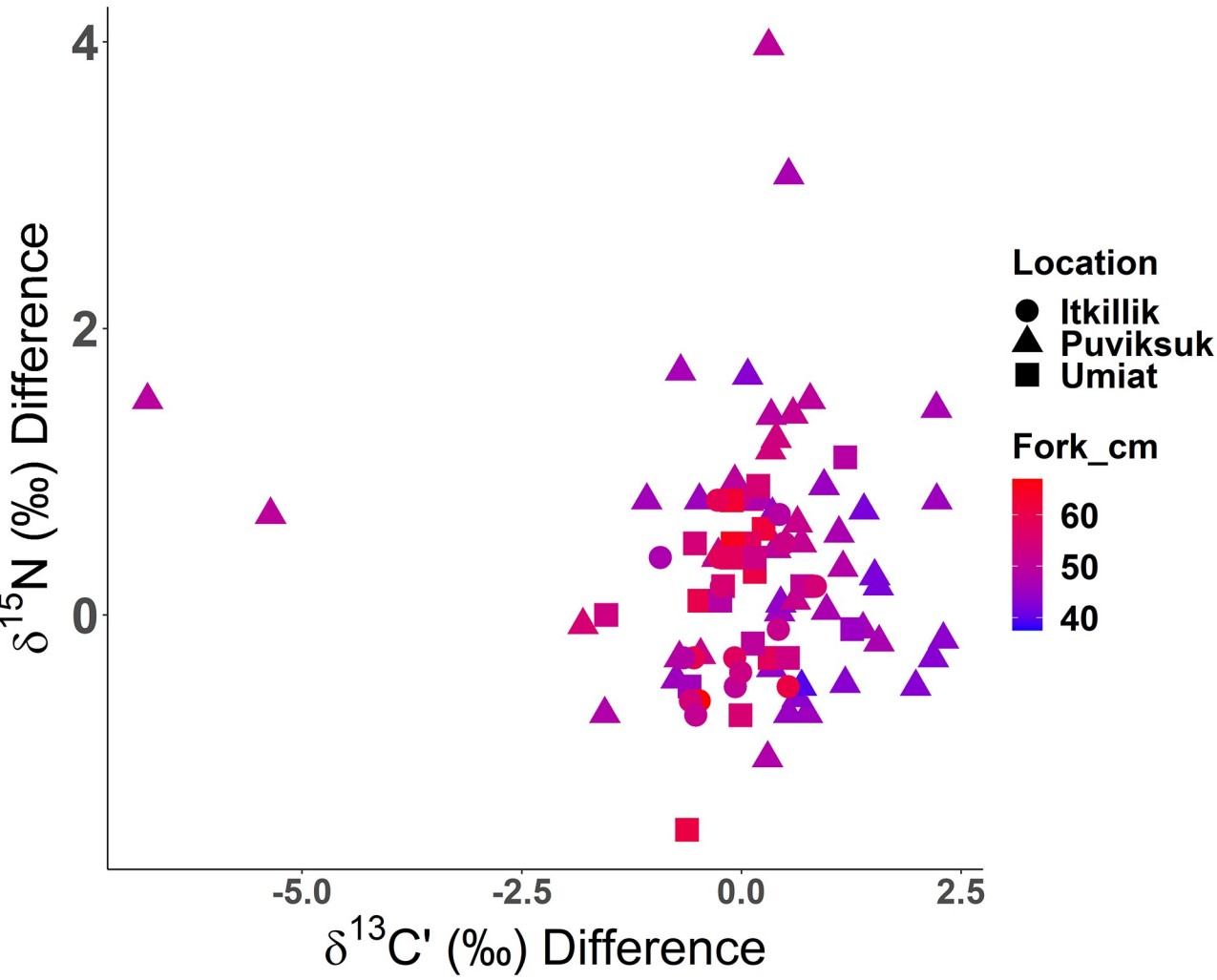

**Fig 3. Isotope tissue differences.** The difference in $\delta^{15}$N and $\delta^{13}$C' (muscle minus liver) for each Broad Whitefish (*Coregonus nasus*) sampled in the Colville River, AK, USA. Scatter plot shapes represent the collection site locations (Itkillik = circle, Puviksuk = triangle, Umiat = square) and the color represents the individual fish's length (blue ≤ 40 cm, purple = 50 cm, red ≥ 60 cm). For both isotope ratios, positive values indicate a shift from foraging food sources from more enriched (e.g., marine gastropod) to a food source with more depleted values (e.g., lacustrine amphipod), while negative values indicate the opposite.

### $\delta^{18}$O and $\delta$D

Muscle $\delta^{18}$O ranged from 4.5 to 10.9‰ with a mean of 7.5‰ (S.D. = 1.33‰; Fig 4) and fit within the range of modeled isotopic values for nearshore marine to inland Arctic regions [29]. Muscle $\delta$D was much more variable, ranging from -237.6 to -158.9‰ with a mean of -191.0‰ (S.D. = 12.89‰; Fig 4).

### Isotopic niche identification

Cluster analysis of muscle $\delta^{13}$C', $\delta^{15}$N, $\delta^{18}$O, and $\delta$D indicated that grouping samples into four levels was best supported by the data (Fig 5A), explained 76.7% of the information on the first two dimensions, and provided good separation of clusters on dimension one (Fig 5B). Fish within cluster group one (n = 55) contained a broad range of $\delta^{13}$C' that were all greater than -27‰, had $\delta^{15}$N between 7 and 10‰, and had $\delta^{18}$O and $\delta$D that overlapped with other groups

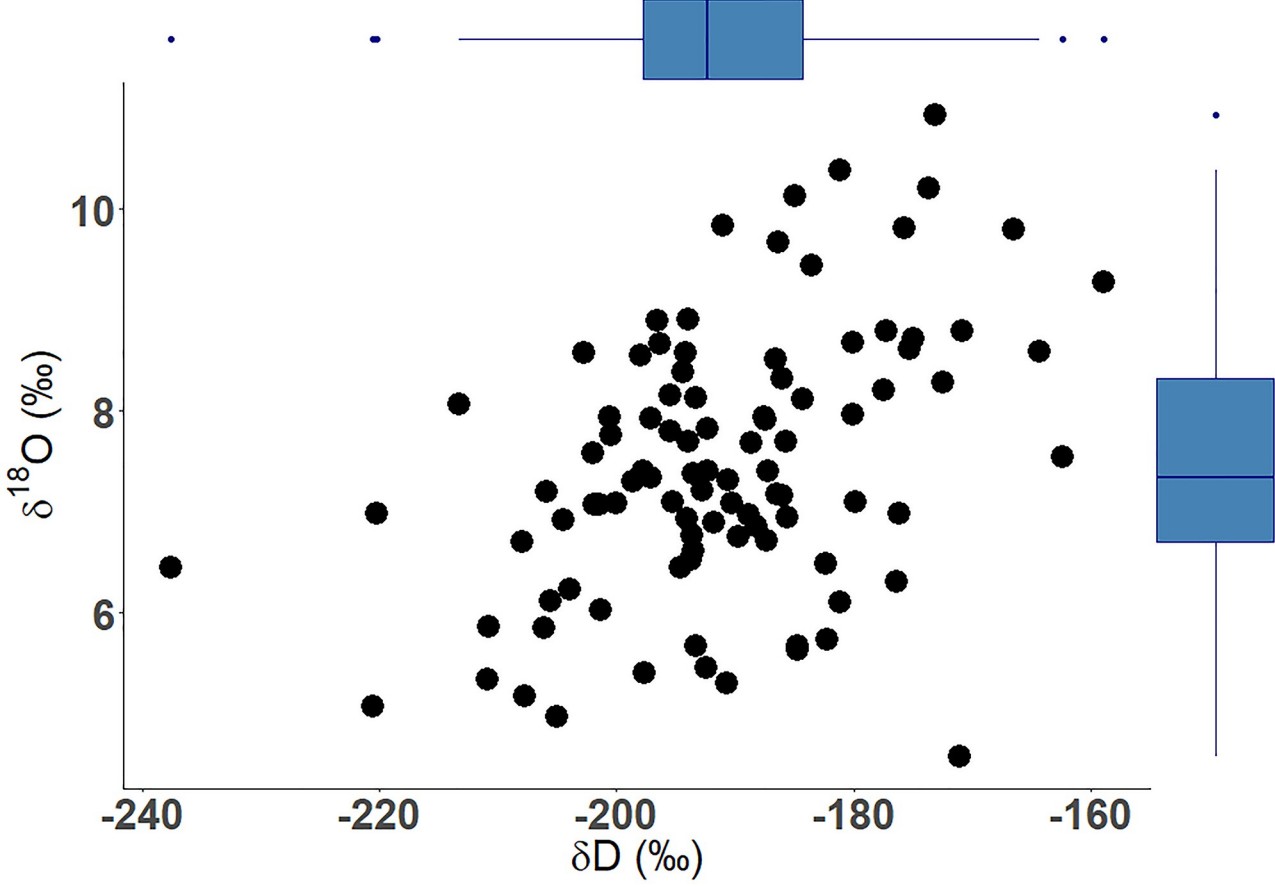

**Fig 4. The scatterplot shows stable isotopes $\delta^{18}O$ versus $\delta D$ along with the associated boxplot for each isotope measured in muscle of Broad Whitefish (*Coregonus nasus*) from the Colville River, AK, USA.** Boxplots show median values (horizontal black line), interquartile range (IQR) (box with blue outline), the maximum value within 1.5 times the IQR (vertical black line), and outside values are greater than 1.5 times the IQR (black dots).

(Fig 6; S4 Table). Individuals within cluster group two (n = 14) contained the most enriched $\delta^{13}C'$, $\delta^{18}O$, and $\delta D$ relative to the other groups (Fig 6; S4 Table). Cluster group three (n = 24) contained a broad range of $\delta^{13}C'$ that were all less than -26.3‰, mean $\delta^{18}O$ that was lower than group 2 and higher than group 1, mean $\delta^{15}N$ that was similar to group 2, and mean $\delta D$ were similar to group 1 (Fig 6; S4 Table). Cluster group four (n = 4) were the most depleted in $\delta^{13}C'$, $\delta^{18}O$, and $\delta D$ but had $\delta^{15}N$ similar to group 1 (Fig 6; S4 Table).

## Variation in isotopic values within life history strategy

Stable isotope values ($\delta^{13}C'$, $\delta^{15}N$, $\delta^{18}O$, $\delta D$) within otolith-derived life history groups (anadromous, semi-anadromous, and nonanadromous) show differences, but also considerable overlap. Nonanadromous individuals (n = 8) had mean $\delta^{13}C'$, $\delta^{18}O$, and $\delta D$ that were depleted compared to semi-anadromous (n = 17) and anadromous (n = 36) individuals (Fig 7; S5 Table). Mean $\delta^{15}N$ was similar for the three life history groups and significant overlap was present (Fig 7B; S5 Table). Anadromous and semi-anadromous individuals had higher mean $\delta D$ than nonanadromous individuals, but there was substantial overlap among groups (Fig 7C; S5 Table).

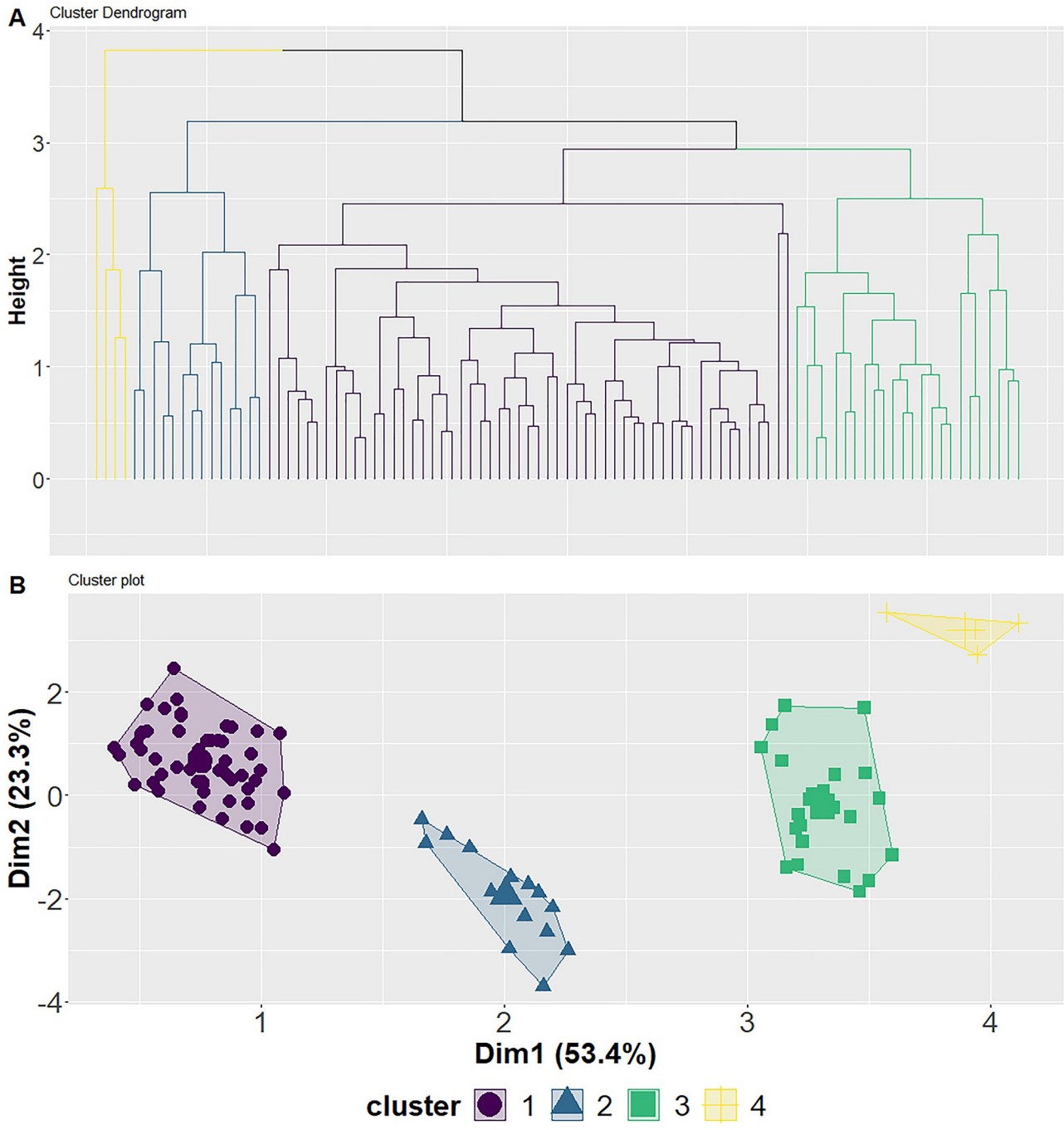

**Fig 5. Hierarchal clustering.** Cluster dendrogram (A) shows the individual Broad Whitefish (*Coregonus nasus*) from the Colville River, AK, USA, clusters (coded by color; purple = cluster 1, blue = cluster 2, turquoise = cluster 3, yellow = cluster 4), and height of the dendrogram. Cluster plot (B) shows each cluster (purple circle = cluster 1, blue triangle = cluster 2, turquoise square = cluster 3, yellow plus = cluster 4) overlaid across two principal components.

## Discussion

Our research revealed that Broad Whitefish utilized numerous isotopic niches and suggests that various successful foraging strategies exist (e.g., residency, migration) to track seasonally

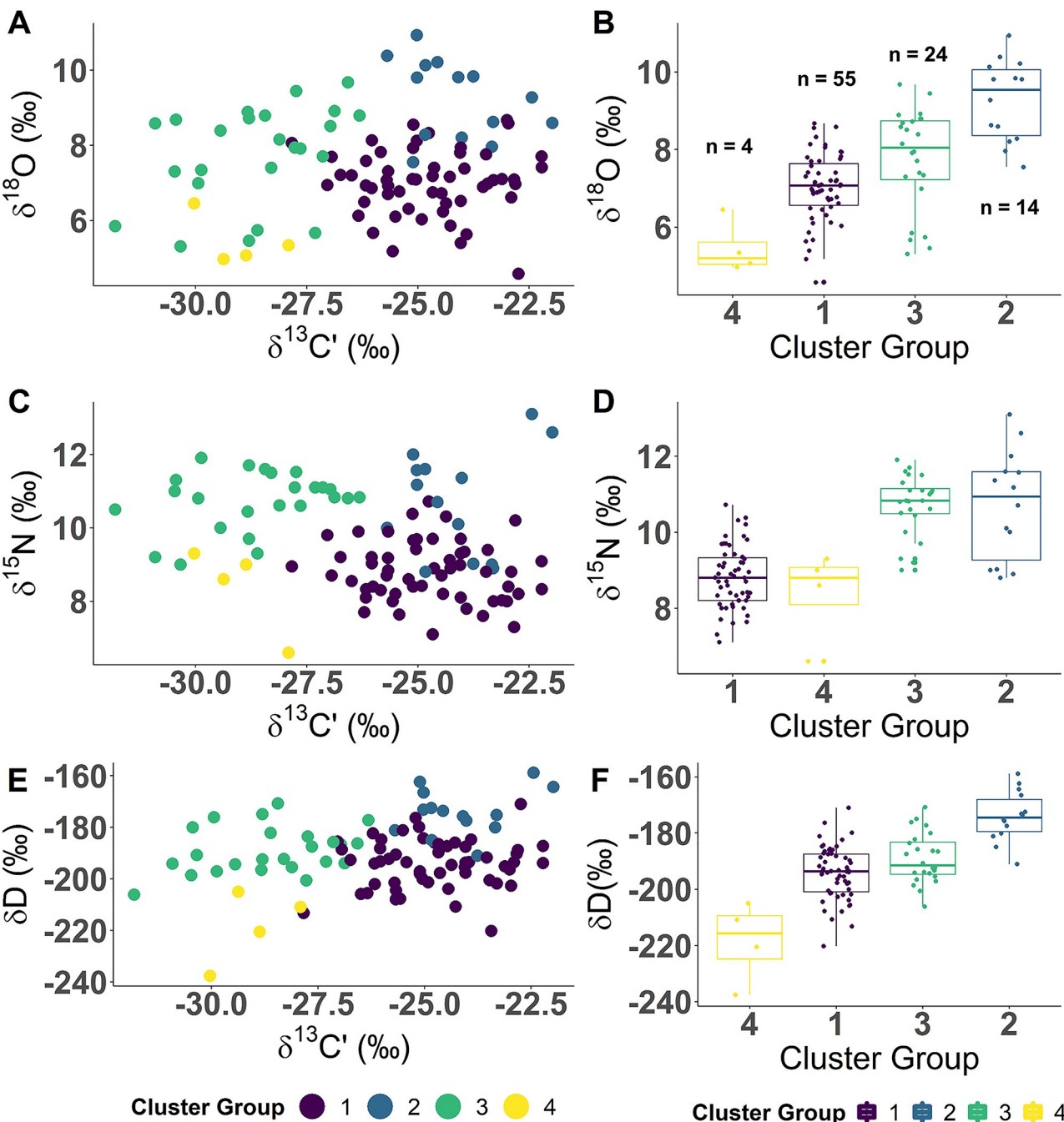

**Fig 6. Isotope values and hierarchal clustering groups.** Scatterplots (A, C, E) show stable isotopes $\delta^{18}O$, $\delta^{15}N$, and $\delta D$ versus $\delta^{13}C'$ along with the associated cluster group (purple = cluster 1, blue = cluster 2, turquoise = cluster 3, yellow = cluster 4) of Broad Whitefish (*Coregonus nasus*) from the Colville River, AK, USA. Boxplots (B, D, F) show stable isotopes $\delta^{18}O$, $\delta^{15}N$, and $\delta D$ by cluster group (purple = cluster 1, blue = cluster 2, turquoise = cluster 3, yellow = cluster 4) along with median (horizontal black line), interquartile range (IQR) (box with colored outline), and maximum value within 1.5 times the IQR (vertical colored line).

available Arctic food resources. The use of a spectrum of freshwater to marine habitats suggests a generalist foraging strategy at the population level, but specialization of foraging habitats where individuals tended to remain in their respective isotopic niches at least for the summer period. Otolith microchemistry demonstrated that Broad Whitefish could switch isotopic

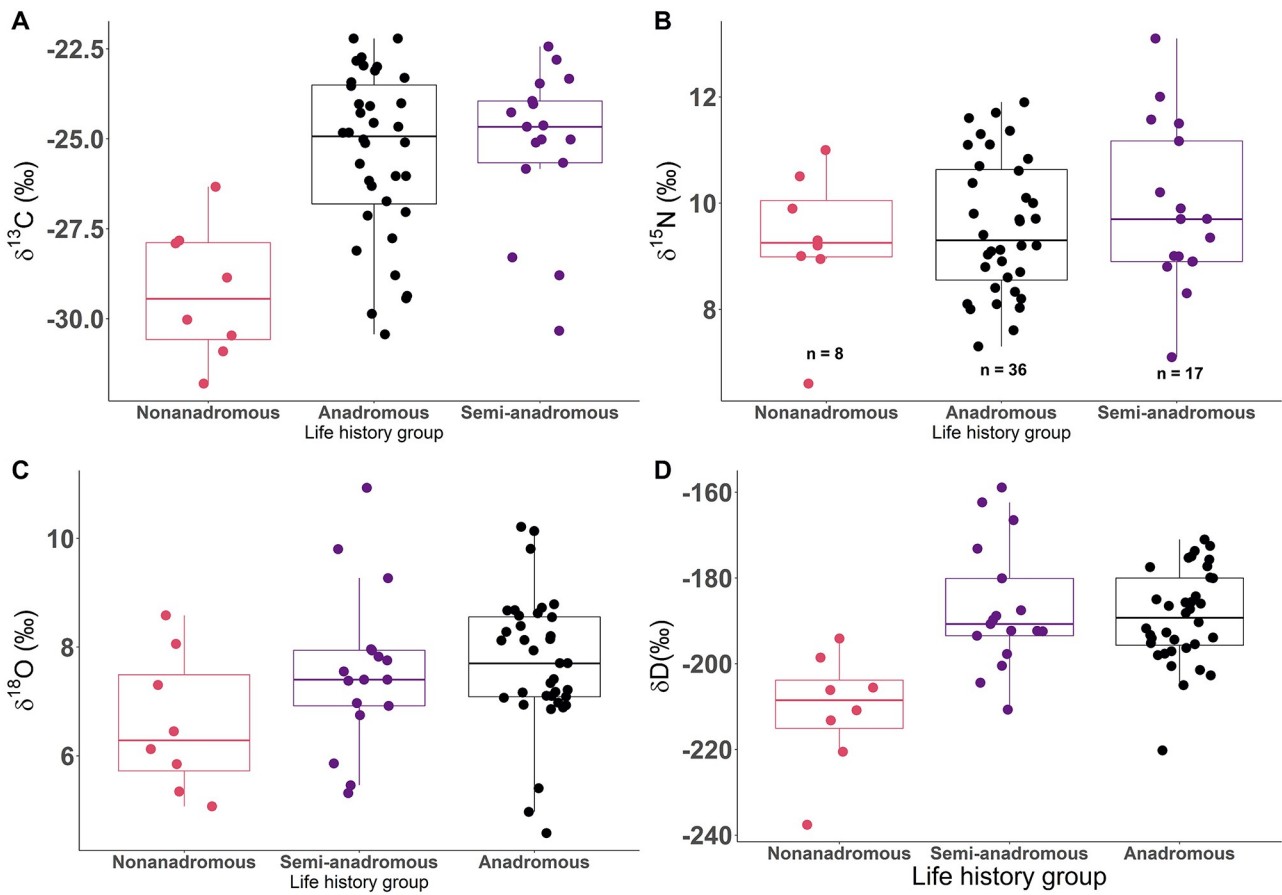

**Fig 7. Isotope values and life history group.** Boxplots (A, B, C, D) showing $\delta^{13}C'$, $\delta^{15}N$, $\delta^{18}O$, and $\delta D$ by Broad Whitefish (*Coregonus nasus*) life history group (anadromous (black), semi-anadromous (purple), nonanadromous (pink), along with median (horizontal black line), interquartile range (IQR) (box with colored outline), and maximum value within 1.5 times the IQR (vertical colored line).

niches, suggesting that prey profitability may change in certain habitats, which could be driven by environmental changes or behavioral fitness decisions. We found otolith microchemistry to be a reliable method to infer Broad Whitefish life history strategies and, when coupled with stable isotope analysis of tissues, provided integrated information on long-term life history patterns, diet, and habitat occupancy for several months prior to capture. This level of diversity and flexibility suggests that the population is ecologically intact and presumably confers some resilience to localized habitat change and disturbance but, as for any fish population, rapid and large-scale landscape changes pose a risk to the long-term stability of Colville River Broad Whitefish.

## Variation in $\delta^{13}C'$ and $\delta^{15}N$ among Broad Whitefish

Broad Whitefish exhibited a large range in $\delta^{13}C'$ and $\delta^{15}N$ within muscle and liver tissue, which suggests fish in this population are consuming food resources across a diversity of habitats and trophic levels. Our data aligns with previous research documenting the importance of freshwater, lagoon, nearshore marine, and marine carbon for anadromous fish food webs [103–105]. Research on Broad Whitefish diet has also revealed that fish are generalist foragers, consuming a variety of benthic and pelagic prey items depending upon age and prey

abundance [12, 27]. In Arctic lentic and lotic ecosystems, food availability can be limited across space and time and generalist foraging strategies, in which individuals feed on a variety of prey items, promote improved survival, rapid growth, and resilience to environmental variability [106]. For example, Broad Whitefish in Arctic Alaska have been shown to consume a variety of pelagic and benthic invertebrate prey items within a single lake, which had similar $\delta^{13}C$ [12]. Conversely, individuals caught in a lake with connection to a stream network exhibited a larger range in $\delta^{13}C'$ and $\delta^{15}N$, suggesting that individuals may be accessing a broader range of prey options [12]. Our results show a slightly larger range of values, supporting the concept of a highly migratory generalist foraging strategy. A flexible foraging strategy in which both benthic and pelagic prey items are utilized enables individuals to efficiently shift between prey items and rapidly accumulate necessary energy reserves prior to a long and cold winter period [107]. Differences among individuals suggest that there is a diversity of foraging specializations, but since we did not analyze stomach contents for each individual we can not draw conclusions about an individuals diet. Therefore, we do not know if individuals are consuming isotopically similar prey items or many prey items with different isotopic values that average to a middle $\delta^{13}C'$ and $\delta^{15}N$ [17].

## Differences in $\delta^{13}C'$ and $\delta^{15}N$ between tissues

Differences in $\delta^{13}C'$ between muscle and liver suggest that most individuals fed on prey with similar $\delta^{13}C$ (+/- 2.5‰) over weeks and months, while a few individuals switched to prey sources or locations with different $\delta^{13}C$. Previous research demonstrates differences in isotope turnover rates between liver and muscle tissue in fish [108–110] due to the association with metabolism rather than growth in liver tissue [108]. Hesslein et al. estimated the half-life of $\delta^{13}C$ and $\delta^{15}N$ to be 101 days in juvenile Broad Whitefish muscle tissue, but due to slow growth in adult fish, it is likely that the turnover rate for muscle tissue could be years [27]. Consequently, the turnover rates remain unknown for Broad Whitefish and our data may reflect the integration of prey resources over longer periods. Interestingly, the two individuals that had larger muscle-liver disparity each had semi-anadromous life histories that may facilitate feeding within habitats with isotopically different $\delta^{15}N$. Combined, these results suggest that multiple generalist-foraging strategies and potentially life histories [82] exist among the population, likely taking advantage of a variety of non-overlapping habitats and variations in the spatial abundance of prey.

## Muscle $\delta^{18}O$ and $\delta D$ and habitat use

The observed range of $\delta^{18}O$ (4.5–10.9‰; Fig 5) suggests that a spectrum of habitat, from low elevation nearshore and estuary habitat to inland higher elevation lakes and rivers, are being utilized by Broad Whitefish. Broad Whitefish $\delta^{18}O$ overlap those of Pink Salmon (*Oncorhynchus gorbuscha*) caught in the Colville River, confirming that some individuals utilize similar nearshore marine habitat leading up to entering freshwater (S3 Table). However, local processes associated with waterbody (e.g., depth, groundwater flow) and origin of water sources (i.e., precipitation, snowmelt, glacier, spring) as well as evaporative effects influence $\delta^{18}O$, which transfer up the food web, further complicating interpretations [96, 111]. Deuterium isotopes are influenced by both isotopic exchange with water during protein syntheses and metabolic water and therefore are a better trophic tracer of aquatic food webs [96]. The large range in $\delta D$ found here (-237.6– -158.9‰) suggests that food is consumed across a range of habitats. Our results show that individuals with nonanadromous life history types had more depleted $\delta D$ compared to anadromous individuals. However, even within individuals that only spent time in freshwater, $\delta D$ had considerable variation, suggesting that some combination of

local environmental water and dietary source effect are likely influencing the variation in values. This is supported by previous research, which documented that water δD had minor influence on chironomid and fish tissue δD [96]. These isotopic fractionation processes make it challenging to assign fine-scale habitat use of Broad Whitefish based on isotopes but are useful in differentiating between certain habitats (e.g., freshwater vs. marine habitats) and aquatic ecosystems that have distinct differences in primary producer energy sources (e.g., clear shallow lakes dominated by algae sources vs. turbid rivers dominated by terrestrial sources).

## Broad Whitefish isotopic niches

Our results show the greatest support for four isotopic niches, but variation within cluster groups suggests a more complex interpretation of physical and biological resources utilized by Broad Whitefish. The hierarchal clustering approach partitioned Broad Whitefish into groups with similar muscle isotopic signatures and provided integrated records of aquatic habitat occupied and prey resources consumed. The isotope cluster groups identified here represent generalized ecological niches utilized by Broad Whitefish over the growing season (ca. three months).

The vast majority of individuals had isotopic signatures representative of coastal river, stream, and lake habitat (Group 3) or coastal lagoon and delta habitat (Group 1) which, as corroborated by previous research (e.g., [12, 27, 112]), signifies the importance of these two ecological niches. Within these cluster groups, it is likely that individuals with more negative values inhabited river deltas that receive significant terrestrial inputs while those with less negative values use coastal lagoon areas and consume prey items that incorporate more marine sources [104]. Cluster group four contained the most depleted $\delta^{13}C'$ and is representative of freshwater food webs where terrestrial (e.g., peat, detritus, soil organic matter) and freshwater carbon (e.g., algae, macrophytes) sources form the base of the food web. Individuals within cluster group four also contained the lowest δD and $\delta^{18}O$, which suggests these fish are using freshwater habitats more inland, and potentially at higher elevations. However, it is also possible that evaporative processes in shallow water bodies are depleting water isotopic values, which then transfer up through the food web [30]. Conversely, individuals within cluster group two had enriched $\delta^{13}C'$, δD, and $\delta^{18}O$ relative to the other groups. These values suggest that individuals within this group spent the majority of their time in nearshore marine areas and consumed prey that primarily utilize marine-based carbon [113].

We suspect that the within-and among-cluster variation was caused by the influence of aquatic habitat heterogeneity that influences stable isotopes within Arctic food webs. Freshwater habitats are influenced by their position in the landscape (e.g., geographic and elevational), physical properties of the waterbody (e.g., morphometry), and biogeochemical processes within the waterbody, which cumulatively influence the isotopic composition of food webs from primary producers up [22, 95]. For example, if a waterbody is shallow and clear, it is likely that autochthonous pathways (e.g., algae) will provide greater support to the food web base [114] and consequently, $\delta^{13}C$ and δD in primary consumers will tend to be more negative compared to sites that depend upon terrestrial peat, or marine-derived carbon inputs [22, 30, 113]. The variation in isotopes suggests that a range of allochthonous and autochthonous carbon from freshwater, terrestrial and marine sources is creating a diversity of food resources with different isotopic values within and between similar waterbody features. For example, the $\delta^{13}C$ can reflect the pelagic-benthic primary production continuum in larger lakes, the terrestrial-aquatic continuum in rivers and riverine lakes, or the freshwater-marine continuum. The range of values could also be caused by a variety of ecological niches utilized by fish, with individuals centered in a cluster more dependent on one specific niche or prey item, while individuals near the periphery may migrate between and utilize multiple niches or switch between

isotopically different prey items, thereby utilizing a portfolio of resources across the Arctic, as seen in other locations [115, 116].

## Life history strategies

Comparing muscle tissue stable isotope ratios within otolith-derived life history strategies revealed high isotopic niche diversity. Otolith microchemistry confirms that individuals with more enriched muscle tissue $\delta^{13}C'$, $\delta D$, and $\delta^{18}O$ tend to use marine habitats (except for over-wintering), while individuals that frequently move between habitats (i.e., freshwater, estuarine, marine) had less enriched values. Both of these patterns are supported by evidence for diverse anadromous and semi-anadromous life history patterns [45, 117]. Numerous individuals with tissue isotopic values reflective of freshwater habitats were revealed by otolith microchemistry to be anadromous or semi-anadromous. These individuals utilized fresh water for months or years prior to capture but had also previously spent significant time in marine habitat as anadromous or semi-anadromous individuals [82]. Therefore, if only tissue samples were used to classify life history strategy, they would have been misclassified. Such abrupt shifts from marine to freshwater habitat use by long-lived anadromous fish has been documented for Coregonids [34, 117, 118]. Conversely, our results show that $\delta^{13}C'$, $\delta D$, and $\delta^{18}O$ are generally predictive of nonanadromous individuals, with $\delta^{13}C'$ and $\delta^{18}O$ reflecting the proportion of time spent in freshwater versus estuarine habitats, with individuals that spent their entire lives in fresh water having the most depleted values.

## Conservation implications

The isotopic niches identified here represent the important habitats utilized by Broad Whitefish across the Beaufort Sea region. The variation within and among isotopic niches suggests that Broad Whitefish utilize a diversity of habitats within freshwater, estuarine, and marine habitats. For example, an individual may exclusively use freshwater lake habitat or move between river, stream, and lake habitats to forage. Diverse foraging behaviors and life history strategies have evolved to maximize foraging efficiency and adapt to dispersed and a shifting heterogeneous mosaic of food resources in the Arctic [12]. Climate change is rapidly altering the Arctic landscape [50–54], causing eutrophication [66] and browning of lakes and rivers [67, 68], altering food web dynamics and potentially reducing fitness for Broad Whitefish that utilize benthic prey items, which could lead to reduced diversity of foraging strategies, slower growth, or lower survival. Arctic riverscapes contain a myriad of stream and lake networks that are at risk from anthropogenic fragmentation that could create barriers (e.g., perched road culverts, drying of channel segments), hindering movement patterns and reducing Broad Whitefish access to food resources. Arctic oil and gas development infrastructure has caused cumulative impacts to permafrost [53, 54], which can cause stream flow modifications that can affect fish access to important habitats. Arctic development fragments and disrupts aquatic ecosystems [119], which can further introduce stressors to juvenile and adult fishes [52, 119] that include increased sedimentation [120–123], modifications of streamflow [124], obstructions to fish passage [125–127], reduced instream habitat quality [128], and pollution [129].

To help buffer populations, it will be necessary for land managers and conservation planners to maintain natural flow regimes, limit barriers along nearshore areas, and preserve aquatic habitat complexity. Preserving connectivity is also important for reducing impacts from infrastructure like roads and from streamflow changes associated with ongoing climate change. Maintaining a diversity of connected niches will facilitate long-term population stability, buffering populations from future environmental and anthropogenic perturbations [130–132].

## Supporting information

**S1 Table. Summary of Broad Whitefish sampled.** Table displaying population structure, tissues sampled, and sample size for Broad Whitefish (*Coregonus nasus*) caught at three locations within the Colville River, AK, USA watershed.
(DOCX)

**S2 Table. Summary of isotope data from Arctic ecosystems used in Fig 2.** Table displaying $\delta^{15}N$ and $\delta^{13}C$ for various Arctic plants invertebrates, fish, and mammals.
(DOCX)

**S3 Table. Summary of isotope data for additional fish species sampled.** Summary of muscle tissue δD and $\delta^{18}O$ for Pink Salmon (*Oncorhynchus gorbuscha*) and Northern Pike (*Esox lucius*) caught in the lower Colville River, AK, USA.
(DOCX)

**S4 Table. Stable isotope ($\delta^{13}C'$, $\delta^{15}N$, $\delta^{18}O$, δD) mean and range values for cluster groups from Broad Whitefish (*Coregonus nasus*) caught in the Colville River, AK, USA.**
(DOCX)

**S5 Table. Stable isotope ($\delta^{13}C'$, $\delta^{15}N$, $\delta^{18}O$, δD) mean and range values for classified life history types of Broad Whitefish (*Coregonus nasus*) caught in the Colville River, AK, USA.**
(DOCX)

**S1 File.**
(XLSX)

## Acknowledgments

We would like to thank the Alaska Cooperative Fish and Wildlife Research Unit (AKCF-WRU) staff, Native Village of Nuiqsut (NVN) Tribal Council staff, U.S. Fish and Wildlife Service (USFWS) Fairbanks Field Office staff, Bureau of Land Management (BLM) Arctic District Office staff, and the Alaska Established Program to Stimulate Competitive Research for assistance with fieldwork and logistics. We thank J. Nukapigak (NVN) for field logistical support and for sharing his traditional knowledge, and M. Lunde (AKCFWRU), and K. Rine (AKCFWRU) for assistance in the field. We thank L. Bryant (BLM) for assistance with summer activity permits and R. Kemnitz (formerly BLM) for logistical help in Umiat. We thank the University of Alaska Anchorage, Environment and Natural Resource Institute (ENRI) Stable Isotope Lab staff, including A. Brownlee (UAA ENRI), for assistance in analyzing tissue samples. We thank the University of Alaska Fairbanks, Alaska Stable Isotope Facility (ASIF) staff, including Mat Wooller (ASIF) and Karen Spaleta (ASIF) for assistance analyzing otolith microchemistry. We thank A. C. Seitz (CFOS, UAF) and J. A. Falke (USGS AKCFWRU) for reviewing early manuscript drafts. We also thank two anonymous reviewers for their constructive comments, which helped us improve the manuscript. Research was conducted under Bureau of Land Management NPR-A permit #FF097006, the Alaska Department of Fish and Game, Fish Resource permit #SF2015-200, and under the University of Alaska Fairbanks Institutional Animal Care and Use Committee Protocol #901048. The findings and conclusions in this article are those of the authors and do not necessarily represent the views of the U.S. Fish and Wildlife Service. Any use of trade, firm, or product names is for descriptive purposes only and does not imply endorsement by the U.S. Government.

## Author Contributions

**Conceptualization:** Jason C. Leppi, Daniel J. Rinella, Mark S. Wipfli, Matthew S. Whitman.

**Data curation:** Jason C. Leppi.

**Formal analysis:** Jason C. Leppi, Daniel J. Rinella, Mark S. Wipfli.

**Funding acquisition:** Mark S. Wipfli, Matthew S. Whitman.

**Investigation:** Jason C. Leppi.

**Methodology:** Jason C. Leppi, Daniel J. Rinella.

**Project administration:** Mark S. Wipfli, Matthew S. Whitman.

**Supervision:** Mark S. Wipfli, Matthew S. Whitman.

**Visualization:** Jason C. Leppi.

**Writing – original draft:** Jason C. Leppi, Daniel J. Rinella.

**Writing – review & editing:** Jason C. Leppi, Daniel J. Rinella, Mark S. Wipfli, Matthew S. Whitman.

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
