## [Decision Letter · Decision Letter 0]

25 Jan 2022

PONE-D-21-39878Broad Whitefish (*Coregonus nasus *) isotopic niches: stable isotopes reveal diverse foraging strategies and habitat use in Arctic AlaskaPLOS ONE

Dear Dr. Leppi,

Thank you for submitting your manuscript to PLOS ONE. After careful consideration, we feel that it has merit but does not fully meet PLOS ONE’s publication criteria as it currently stands. Therefore, we invite you to submit a revised version of the manuscript that addresses the points raised during the review process.

Specifically, Even though both reviewers found the study interesting, they highlighted a number of important flaws, to the point that one of them suggested a rejection. I have now personally reviewed the ms, and my opinon is that even though it deserves a radical restructuration, it can be considered further for publication in PLOS ONE if the authors manage to address all the points raised by the reviewers, with special attention given to the criticisms made by rev#2.

We look forward to receiving your revised manuscript.

Kind regards,

Giorgio Mancinelli, Ph.D.

Academic Editor

PLOS ONE

Journal Requirements:

2. We note that you have referenced (J.C. Leppi unpublished data) which has currently not yet been accepted for publication. Please remove this from your References and amend this to state in the body of your manuscript: (ie “J.C. Leppi unpublished data”) as detailed online in our guide for authors

Reviewers' comments:

Reviewer's Responses to Questions

**Comments to the Author**

1. Is the manuscript technically sound, and do the data support the conclusions?

Reviewer #1: Yes

Reviewer #2: Partly

2. Has the statistical analysis been performed appropriately and rigorously? 

Reviewer #1: Yes

Reviewer #2: Yes

3. Have the authors made all data underlying the findings in their manuscript fully available?

Reviewer #1: Yes

Reviewer #2: Yes

4. Is the manuscript presented in an intelligible fashion and written in standard English?

Reviewer #1: Yes

Reviewer #2: Yes

5. Review Comments to the Author

Reviewer #1: Thank you for the opportunity to review this interesting work. I found it to be of much potential, yet I have several points of concern that ought to be addressed.

This centers mostly around the introductions structure, a lack of species specific information in the discussion and some further details being required. Please note that I am no expert on the species for which reason my comments on the discussion remained very poor. References have not been checked and I do not judge the language as I am not a native English speaker.

Abstract:

I have no major comments to the abstract except one suggestion: While the findings are presented fairly well, the implications are handled only with one minor half-sentence statement at the end “maintaining these diverse connected habitats will facilitate long term population stability, buffering populations from future environmental and anthropogenic perturbations.”. Perhaps the authors could expand slightly on what this (perhaps by condensing the findings if word limit is an issue).

Introduction:

- Line 57-67: I haven’t read such a broad yet very appealing and nice beginning of an introduction in a while.

- The introduction is currently excessively long and hard to follow as the nature of the study (isotopes) is somewhat minimalized due to the large and overwhelming sections on the species, the Arctic, and oil/gas development.

- I would recommend moving the section on the Arctic Alaska prior to the introduction of the broad whitefish and do you really need the excessive introduction on the oil and gas development? I would suggest combining (and streamlining) the section on Alaska and oil/gas development and moving it up. This way, the section on the target species would be followed by what currently starts in line 146: “Broad Whitefish population…”.

Methods:

- Line 207-215: Perhaps adding coordinates would be a welcomed detail.

- Line 215: Was this euthanization approach in line with regional / local guidelines? Please add information on the underlying legislation that permits this procedure as I cant see from the listed Permit that this was included. Adding a sentence if this approach was in lines with said guidelines or laws (if so) should suffice.

- How was the sex identified?

- Why fork and not total length?

- Line 231-268: I actually enjoyed this introduction to stable isotopes.

- Line 269-271: Could you perhaps add a reference?

- Line 271-287: I concur completely with the argumentation

- Line 311-326: Although I worked with isotopes and niche space myself and quite excessively, I see this section very hard to follow and ultimately, to understand. Perhaps the authors could build on the already written text and try to better explain it?

- Line 327-329: Question 1: More generally, I wonder why didn’t you perform actual diet analysis?

- Line 331-332: Please add a reasoning for that analysis.

Results:

Overall, the presentation of the results is very orderly and condensed.

- Generally, I am concerned about the origin on these species’ data. Although they are listed in Table S2, its questionable how the data was made comparable. Perhaps I overlooked it previously, but comparability among sites / samples from different period can be difficult.

Discussion:

A general comment: In order to review the discussion I would need a relatively more profound knowledge on the ecology of the target species – which I don’t have. Hence, I am not able to look in depth into behavioural or conservation implications. Yet, this might be something the authors would want to provide the reader with.

- Line 472-474: I would suggest putting more emphasize on the differences between dN/dC and all analysed istopes in conjunction into this first section. This high resolution obtained and the identified differences are crucial and a neat find as well. Even if just briefly and later on discussed in detail

- Line 506-514: One crucial point that has been neglected is the physiological difference between those tissues and the associated bioaccumulation factors of them.

- Line 599: Is there any actual measure that could be undertaken aside from conserving the current situation?

Figure 1: This is a highly interesting figure. I have some problems with the direct comparability of species like Arctic Char and Northern Pike due to the differences in ecosystems, but this figure does an incredibly good job at displaying the huge niche of whitefish.

Figure 2: no comment

Figure 3: no comment

Figure 4: I would suggest not using yellow as it is very hard to see.

Figure 5: I would suggest not using yellow as it is very hard to see.

Figure 6: no comment

Reviewer #2: General comments:

This study examined trophic and life history diversity in a population of broad whitefish inhabiting the Colville River drainage basin in northern Alaska. Approximately 100 adult fish were sampled from the mainstem river and analyzed for C, N, O and H stable isotope compositions in muscle and liver, and for Sr isotopic composition in otoliths. The data were used to interpret diversity in habitat use and trophic ecology within the population. The manuscript is suitable subject matter for PLoS One.

The description of the study is straightforward, and the manuscript is generally easy to read. However, the writing is sometimes too detailed or repetitive and the manuscript could be condensed considerably. I also had some difficulty understanding the data analysis methods, and consequently, with interpreting results. Details are outlined below. Though both the isotopic clustering and otolith chemistry categorization point to three or four life history groupings, they do not appear to correspond strongly. As a result, the interpretation descends into considerable arm-waving that needs to be condensed.

Provided these problems can be dealt with, I think there is sufficient material here for an interesting story to be told. I recommend that the manuscript be rejected in its current form but a re-analyzed and revised version could be reconsidered.

Specific comments and suggestions for revision:

Lines 91-92. Should read “….isotope ratios in animal tissues change in predictable ways across landscapes [29], and when used in combination….”

Lines 101-103. Be more specific here; what does a high 87Sr/86Sr ratio signify? Does “Sr values” mean “Sr ratios” in this context (line 103).

Lines 112-115. Are you suggesting that a short growing season promotes high diversity in habitat and resource use?

Lines 122-145. The information on climate and developmental stressors on the landscape can be condensed into a single paragraph.

Line 153. When you say “rivers across the Beaufort Coastal Plain” are you referring to rivers that are tributary to the Colville (i.e., within the Colville drainage basin) or other nearby rivers that drain directly into the Beaufort Sea?

Line 207. Should read “adult Broad Whitefish”

Line 218. From where on the body was the muscle biopsy removed?

Lines 220-222. Not sure what ‘integration’ implies here exactly. Because these are large, mature and slow-growing fish, I suspect the isotopic change for these tissues is quite slow. Presumably ‘integration’ is primarily through anabolism (rather than tissue replacement) and it seems doubtful that these fish would increase in mass by any more than 20-30% per year.

Lines 234-235. Delete “…and organic carbon from plant detritus and soil”. Organic carbon is part of DOC, not DIC

Line 235. Delete “(fractionation)”. Uptake of carbon dioxide by plants is not fractionation, though fractionation does occur during uptake.

Lines 249-254. These sentences are confusing. The explanation as to why N isotope ratios of DIN may differ between marine and freshwater ecosystems is not clear.

Line 255. Delete “level”

Lines 231-287. This whole section is too detailed for the Methods. Some of it can go in the Introduction, as justification for the isotopic approach, but most of this should be condensed to a single paragraph in the Methods that summarizes how you intend to interpret variation in each of the isotope ratios examined.

Lines 297-306. Again, condense, or move to the supplemental information file.

Lines 319-321. Not clear how the lipid normalization was carried out. Presumably this was only for the C stable isotope ratios?

Lines 328-329. Should read “…between muscle and liver tissues (i.e., muscle minus liver) for both delta15N and delta13C”

Lines 329-330. A correlation analysis of what? Be specific.

Lines 331-334. Not clear why both correlation analysis and regression analysis are carried out on the same data set.

Lines 338-339. Omit “…due to instrument constraints (i.e., time, funding, instrument availability)”. You do not need to justify why not all the otoliths were analyzed.

Lines 339-340. At least briefly mention the otolith prep and the instrumentation used. Presumably LA-ICP-MS?

Line 350. Should read “If Sr data were highly…” Data is plural.

Lines 354-355. Need to define FEB on first usage.

Line 357. What does “[v]” mean in this context?

Lines 352-366. I found this whole paragraph very difficult to understand, right from the calculation of concentrations through to the categorizations.

Lines 380-384, Fig. 2 caption. Because broad whitefish data from this study are being compared with data from other species and studies, the geographic scope of the other data needs to be defined in the caption. Did all the other data come from the same area of northern Alaska? Also, the broad whitefish point should be a mean +/- SD so that its variability is directly comparable to the other points. Using ranges inflates the relative variability.

Lines 390-391. This is poorly worded. Presumably you are talking about relationships between delta13C and delta15N within each of the two tissues? Were the relationships positive or negative?

Line 401, Fig 3 caption. Suggest indicating that blue is less than or equal to 40 cm and red is greater than or equal to 65 cm. Also, should read “For both isotope ratios, positive values indicate…”

Lines 431-435. Table 1 provides the same information as Figure 6. I suggest deleting the table.

Lines 460-463. Table 2 provides the same information as Figure 7. I suggest deleting the table.

Somewhere in Discussion. Should mention how trophic diversity is measured and how this can influence the interpretation of results. Trophic diversity has both within-individual and among-individual components. In this study, diversity is measured among-individuals but not within-individuals. The among-individual diversity is sometimes interpreted as a measure of individual specialization.

Somewhere in Discussion. It should be stated more clearly that delta13C variation can be interpreted in various ways. Delta13C can reflect the pelagic-benthic primary production continuum in larger lakes, and can also reflect the terrestrial-aquatic (or allochthonous-authochthonous) continuum in rivers, or more riverine lakes. Most terrestrial primary production is very 13C depleted, similar to pelagic production. It can also reflect the freshwater-marine continuum, as in the Colville system.

Lines 646-670, Acknowledgements. Condense considerably.

6. PLOS authors have the option to publish the peer review history of their article (what does this mean?). If published, this will include your full peer review and any attached files.

Reviewer #1: No

Reviewer #2: No

---

## [Author Response · Author response to Decision Letter 0]

28 Apr 2022

View Letter

Date: Jan 25 2022 06:28PM

To: "Jason C. Leppi" jcleppi@alaska.edu;jason_leppi@tws.org

From: "PLOS ONE" plosone@plos.org

Subject: PLOS ONE Decision: Revision required [PONE-D-21-39878]

PONE-D-21-39878

Broad Whitefish (Coregonus nasus ) isotopic niches: stable isotopes reveal diverse foraging strategies and habitat use in Arctic Alaska

PLOS ONE

Dear Dr. Leppi,

Thank you for submitting your manuscript to PLOS ONE. After careful consideration, we feel that it has merit but does not fully meet PLOS ONE’s publication criteria as it currently stands. Therefore, we invite you to submit a revised version of the manuscript that addresses the points raised during the review process.

Specifically, Even though both reviewers found the study interesting, they highlighted a number of important flaws, to the point that one of them suggested a rejection. I have now personally reviewed the ms, and my opinon is that even though it deserves a radical restructuration, it can be considered further for publication in PLOS ONE if the authors manage to address all the points raised by the reviewers, with special attention given to the criticisms made by rev#2.

We look forward to receiving your revised manuscript.

Kind regards,

Giorgio Mancinelli, Ph.D.

Academic Editor

PLOS ONE

We thank the academic editor for their review of our manuscript. Your suggestions helped us to greatly improve our manuscript. 

Sincerely, 

Jason Leppi, Dan Rinella, Mark Wipfli, and Matthew Whitman

Note: All line numbers referenced in this document refer to the Revised Manuscript with Track Changes document 

Journal Requirements:

Thank you for the feedback. We reviewed the style requirements and have updated the file names, title page, and supporting information material. 

2. We note that you have referenced (J.C. Leppi unpublished data) which has currently not yet been accepted for publication. Please remove this from your References and amend this to state in the body of your manuscript: (ie “J.C. Leppi unpublished data”) as detailed online in our guide for authors

We removed the unpublished data citation and added a new citation. See L. 647.

Thank you for the feedback we have created a new figure that only contains public domain data layers. See revised figure 1. 

Reviewers' comments:

Reviewer's Responses to Questions

Comments to the Author

1. Is the manuscript technically sound, and do the data support the conclusions?

Reviewer #1: Yes

Reviewer #2: Partly

2. Has the statistical analysis been performed appropriately and rigorously?

Reviewer #1: Yes

Reviewer #2: Yes

3. Have the authors made all data underlying the findings in their manuscript fully available?

Reviewer #1: Yes

Reviewer #2: Yes

4. Is the manuscript presented in an intelligible fashion and written in standard English?

Reviewer #1: Yes

Reviewer #2: Yes

5. Review Comments to the Author

Reviewer #1: Thank you for the opportunity to review this interesting work. I found it to be of much potential, yet I have several points of concern that ought to be addressed.

This centers mostly around the introductions structure, a lack of species specific information in the discussion and some further details being required. Please note that I am no expert on the species for which reason my comments on the discussion remained very poor. References have not been checked and I do not judge the language as I am not a native English speaker.

We thank you for the thoughtful and detailed review of our manuscript. Your comments and suggestions helped us to greatly improve our manuscript. Thank you for taking the time to review our manuscript. 

Sincerely, 

Jason Leppi, Dan Rinella, Mark Wipfli, and Matthew Whitman 

Note: All line numbers referenced in this document refer to the Revised Manuscript with Track Changes document 

Abstract:

I have no major comments to the abstract except one suggestion: While the findings are presented fairly well, the implications are handled only with one minor half-sentence statement at the end “maintaining these diverse connected habitats will facilitate long term population stability, buffering populations from future environmental and anthropogenic perturbations.”. Perhaps the authors could expand slightly on what this (perhaps by condensing the findings if word limit is an issue).

Made change. See L. 53¬¬¬¬¬¬–55.

Introduction:

- Line 57-67: I haven’t read such a broad yet very appealing and nice beginning of an introduction in a while.

Thank you for the positive feedback regarding our introduction. We spent significant time crafting it. 

- The introduction is currently excessively long and hard to follow as the nature of the study (isotopes) is somewhat minimalized due to the large and overwhelming sections on the species, the Arctic, and oil/gas development.

Thank you for the suggestion. We removed the section on Arctic oil and gas development from the introduction. However, we think it is important to keep information on the species (Broad Whitefish) in the introduction because it sets the stage for the research. 

- I would recommend moving the section on the Arctic Alaska prior to the introduction of the broad whitefish and do you really need the excessive introduction on the oil and gas development? I would suggest combining (and streamlining) the section on Alaska and oil/gas development and moving it up. This way, the section on the target species would be followed by what currently starts in line 146: “Broad Whitefish population…”.

Thank you for the suggestion. We removed the section on Arctic oil and gas development from the introduction and moved the paragraph on climate impacts prior to the paragraph that describes Broad Whitefish. See L. 109–121.

Methods:

- Line 207-215: Perhaps adding coordinates would be a welcomed detail.

The location of our sampling occurred at three locations (Puviksuk, Umiat, Itkillik) on the Colville River which are shown in Fig1. We feel as thought the Fig1 maps provide adequate detail for the research.

- Line 215: Was this euthanization approach in line with regional / local guidelines? Please add information on the underlying legislation that permits this procedure as I cant see from the listed Permit that this was included. Adding a sentence if this approach was in lines with said guidelines or laws (if so) should suffice.

Yes, our methods were in line with Institutional Animal Care and Use Committee Protocol standard euthanization procedures to minimize animal suffering. We added a clarifying sentence. See L. 247–248.

- How was the sex identified?

Sex for all individuals was determined visually while taking tissue samples for stable isotope analysis. Females had eggs, males had milt, and a small percentage had neither and were classified as unidentified.

- Why fork and not total length?

Fork length was measured to be more consistent with measurements between individuals. 

- Line 231-268: I actually enjoyed this introduction to stable isotopes.

Thank you, but unfortunately, reviewer #2 requested that we remove this introductory content from the methods section. The vast majority of this content has been removed.

- Line 269-271: Could you perhaps add a reference?

Added reference. See L. 290. 

- Line 271-287: I concur completely with the argumentation

Thank you for your feedback.

- Line 311-326: Although I worked with isotopes and niche space myself and quite excessively, I see this section very hard to follow and ultimately, to understand. Perhaps the authors could build on the already written text and try to better explain it?

Thank you for the suggestion. We updated and rearranged the text to be clearer. We hope that it is now more clear. See L. 332–354.

- Line 327-329: Question 1: More generally, I wonder why didn’t you perform actual diet analysis?

Good question. We did not perform a diet analysis because the individuals caught had empty stomachs. We checked their stomach contents and didn’t find any prey items. The vast majority of individuals were prespawners and it is thought that fish stop or reduce feeding while migrating.

- Line 331-332: Please add a reasoning for that analysis.

Made change. See L. 360–361. 

Results:

Overall, the presentation of the results is very orderly and condensed.

Thank you, we also feel that the results section is presented well.

- Generally, I am concerned about the origin on these species’ data. Although they are listed in Table S2, its questionable how the data was made comparable. Perhaps I overlooked it previously, but comparability among sites / samples from different period can be difficult.

Thank you for the feedback. The goal of the figure is to show a depiction of the variation of � 13C and � 15N across freshwater, estuarine, and marine ecosystems as it relates to our Broad Whitefish samples. Broad Whitefish are highly migratory and likely use vast areas of the Arctic to fulfill their life cycle. The samples that were collected outside the study area are only used to help visualize how Broad Whitefish isotope values relate to other species within Arctic ecosystems. In addition to whitefish samples, we collected samples from Northern Pike and Pink Salmon caught in our nets and invertebrates from several sites in the Colville River watershed. The data and references for species collected is clearly listed in S2 Table.

Discussion:

A general comment: In order to review the discussion I would need a relatively more profound knowledge on the ecology of the target species – which I don’t have. Hence, I am not able to look in depth into behavioural or conservation implications. Yet, this might be something the authors would want to provide the reader with.

Thank you for the feedback. We provide text in the discussion that describes behavioral and conservation implications. See Conservation implications section L. 656–681. 

- Line 472-474: I would suggest putting more emphasize on the differences between dN/dC and all analysed istopes in conjunction into this first section. This high resolution obtained and the identified differences are crucial and a neat find as well. Even if just briefly and later on discussed in detail

Thank you for the feedback. We discuss the difference between δ13Cˈand δ15N in the discussion section labeled “Variation in δ13Cˈand δ15N among Broad Whitefish”. See L. 531–552. 

- Line 506-514: One crucial point that has been neglected is the physiological difference between those tissues and the associated bioaccumulation factors of them.

We updated the text to briefly expand upon the physiological differences between liver and muscle tissue. See L. 557–563. 

- Line 599: Is there any actual measure that could be undertaken aside from conserving the current situation?

Good question. Given our limited knowledge of Arctic Broad Whitefish movement and habitat use and the generally pristine nature of the landscape we think that conserving habitat and minimizing impacts is the right level of detail for this section at this point.

Figure 1: This is a highly interesting figure. I have some problems with the direct comparability of species like Arctic Char and Northern Pike due to the differences in ecosystems, but this figure does an incredibly good job at displaying the huge niche of whitefish.

Thank you. We also feel that Fig 2 (not Fig 1) does a good job at show differences of δ13Cˈand δ15N between ecosystems and across species.

Figure 2: no comment

Figure 3: no comment

Figure 4: I would suggest not using yellow as it is very hard to see.

We think that the colors used in the plot are adequate, which are trying to balance visual appeal with functionality. 

Figure 5: I would suggest not using yellow as it is very hard to see.

We think that the colors used in the plot are adequate which are trying to balance visual appeal with functionality. 

Figure 6: no comment

Reviewer #2: General comments:

This study examined trophic and life history diversity in a population of broad whitefish inhabiting the Colville River drainage basin in northern Alaska. Approximately 100 adult fish were sampled from the mainstem river and analyzed for C, N, O and H stable isotope compositions in muscle and liver, and for Sr isotopic composition in otoliths. The data were used to interpret diversity in habitat use and trophic ecology within the population. The manuscript is suitable subject matter for PLoS One.

The description of the study is straightforward, and the manuscript is generally easy to read. However, the writing is sometimes too detailed or repetitive and the manuscript could be condensed considerably. I also had some difficulty understanding the data analysis methods, and consequently, with interpreting results. Details are outlined below. Though both the isotopic clustering and otolith chemistry categorization point to three or four life history groupings, they do not appear to correspond strongly. As a result, the interpretation descends into considerable arm-waving that needs to be condensed.

Provided these problems can be dealt with, I think there is sufficient material here for an interesting story to be told. I recommend that the manuscript be rejected in its current form but a re-analyzed and revised version could be reconsidered.

We thank you for the thoughtful and detailed review of our manuscript. Your comments and suggestions helped us to greatly improve our manuscript. Thank you for taking the time to review our manuscript. 

Sincerely, 

Jason Leppi, Dan Rinella, Mark Wipfli, and Matthew Whitman 

Note: All line numbers referenced in this document refer to the Revised Manuscript with Track Changes document 

Specific comments and suggestions for revision:

Lines 91-92. Should read “….isotope ratios in animal tissues change in predictable ways across landscapes [29], and when used in combination….”

Made change. See L. 93–95.

Lines 101-103. Be more specific here; what does a high 87Sr/86Sr ratio signify? Does “Sr values” mean “Sr ratios” in this context (line 103).

Made change. See L. 106.

Lines 112-115. Are you suggesting that a short growing season promotes high diversity in habitat and resource use?

No, we don’t think it has to do with the length of the growing season, but rather the availability, timing, and duration of aquatic habitats. We suspect that surface water phenology influences Broad Whitefish foraging ecology similar to other species that live in ecosystems where the availability of habitat and food resources frequently change. See reference below. 

Heim KC, McMahon TE, Calle L, Wipfli MS, Falke JA. A general model of temporary aquatic habitat use: water phenology as a life history filter. Fish Fish. 2019;20:802–816. https://doi.org/10.1111/faf.12386

Lines 122-145. The information on climate and developmental stressors on the landscape can be condensed into a single paragraph.

Made change. Removed paragraph on development stressors from the introduction. 

Line 153. When you say “rivers across the Beaufort Coastal Plain” are you referring to rivers that are tributary to the Colville (i.e., within the Colville drainage basin) or other nearby rivers that drain directly into the Beaufort Sea?

Good question. We don’t know where fish are foraging, but it is likely from a variety of habitats across the Arctic. Broad Whitefish have been observed using a variety of freshwater habitats within the Beaufort Coastal plain as well as estuarine and nearshore habitat along the Beaufort Sea coast. There is evidence that Broad Whitefish migrate from nearby rivers such as Fish Creek or Little Putuligayak River to the Colville to potentially spawn and/or overwinter (Morris 2000;2003), but fish could utilize other areas. There is also documented evidence of Broad Whitefish using nearshore areas to the east of the Colville River during the summer. 

Morris WA. Seasonal movements of broad whitefish (Coregonus nasus) in the freshwater systems of the Prudhoe Bay oil field. 2000. [Master's thesis, University of Alaska Fairbanks]. ScholarWorks@UA publishing. http://hdl.handle.net/11122/6783

Morris WA. Seasonal movements and habitat use of Arctic grayling (Thymallus arcticus), burbot (Lota lota), and broad whitefish (Coregonus nasus) within the Fish Creek drainage of the National Petroleum Reserve-Alaska, 2001–2002; Technical Report No. 03-02. 2003. Alaska Department of Natural Resources, Office of Habitat Management. https://www.adfg.alaska.gov/static/home/library/pdfs/habitat/03_02.pdf

Green DG, Priest JT, Gatt KP, Sutton TM. Beaufort Sea Nearshore Fish Monitoring Study: 2018 Annual Report. Report for Hilcorp Alaska, LLC by the University of Alaska Fairbanks, College of Fisheries and Ocean Sciences, Department of Fisheries, Fairbanks, Alaska.2018. http://www.north-slope.org/assets/images/uploads/2018_Beaufort_Sea_Nearshore_Fish_Monitoring_Annual_Report.pdf

Line 207. Should read “adult Broad Whitefish”

Made change. See L. 223.

Line 218. From where on the body was the muscle biopsy removed?

From the epaxial muscle near the dorsal fin. We added clarifying text to the manuscript. See L. 234.

Lines 220-222. Not sure what ‘integration’ implies here exactly. Because these are large, mature and slow-growing fish, I suspect the isotopic change for these tissues is quite slow. Presumably ‘integration’ is primarily through anabolism (rather than tissue replacement) and it seems doubtful that these fish would increase in mass by any more than 20-30% per year.

Yes, we agree and have incorporated your suggestion into the manuscript. See L.236 and L. 238–241. It is likely that isotopic change of Broad Whitefish muscle is much longer that the stated 88 days, but we really don’t know how long. Vander Zanden et al. 2015 provide a good review on the topic and evidence that the half-life of isotopes in fish muscle tissue is around 88 days. Within the review, Hesslein et al. 1993 provide evidence for Broad Whitefish muscle half-life of 101.9 days. 

Vander Zanden MJ, Clayton MK, Moody EK, Solomon CT, Weidel BC. Stable isotope turnover and half-life in animal tissues: a literature synthesis. PLOS ONE. 2015;10 (1): e0116182. https://doi.org/10.1371/journal.pone.0116182

Hesslein RH, Hallard KA, Ramlal P. Replacement of sulfur, carbon, and nitrogen in tissue of growing broad whitefish (Coregonus nasus) in response to a change in diet traced by delta S-34, delta C-13, and delta N-15. Can J Fish Aquat Sci. 1993; 50:2071-2076. https://doi.org/10.1139/f93-230

Lines 234-235. Delete “…and organic carbon from plant detritus and soil”. Organic carbon is part of DOC, not DIC

Deleted text

Line 235. Delete “(fractionation)”. Uptake of carbon dioxide by plants is not fractionation, though fractionation does occur during uptake.

Deleted text

Lines 249-254. These sentences are confusing. The explanation as to why N isotope ratios of DIN may differ between marine and freshwater ecosystems is not clear.

Deleted text

Line 255. Delete “level”

Deleted text

Lines 231-287. This whole section is too detailed for the Methods. Some of it can go in the Introduction, as justification for the isotopic approach, but most of this should be condensed to a single paragraph in the Methods that summarizes how you intend to interpret variation in each of the isotope ratios examined.

Good suggestion. We reduced the method section significantly by removing text. Deleted text, see revised methods section, L. 251–288 and L. 297–308. 

Lines 297-306. Again, condense, or move to the supplemental information file.

OK. We condensed the text by removing the description on internal standards. See L. 319–325.

Lines 319-321. Not clear how the lipid normalization was carried out. Presumably this was only for the C stable isotope ratios?

Yes, we conducted lipid normalization only for the δ13C values. We revised the text to be more clear. See L. 332–338. 

Lines 328-329. Should read “…between muscle and liver tissues (i.e., muscle minus liver) for both delta15N and delta13C”

Made change. See L. 357.

Lines 329-330. A correlation analysis of what? Be specific.

This information is within the first paragraph. See L.355–357. We also added clarifying text at L. 360–361.

Lines 331-334. Not clear why both correlation analysis and regression analysis are carried out on the same data set.

To help determine if an individual's diet remained stable or changed over the summer period, we visually and quantitatively compared the difference between δ13Cˈand δ15N muscle and liver tissues (i.e., muscle minus liver) for both δ13Cˈand δ15N using multiple approaches. See L. 355–363.

Lines 338-339. Omit “…due to instrument constraints (i.e., time, funding, instrument availability)”. You do not need to justify why not all the otoliths were analyzed.

Made change. Deleted text.

Lines 339-340. At least briefly mention the otolith prep and the instrumentation used. Presumably LA-ICP-MS?

OK. Made change. See L. 368–373.

Line 350. Should read “If Sr data were highly…” Data is plural.

Made change. See L. 389.

Lines 354-355. Need to define FEB on first usage.

Thank you. We updated the reference material name in the text. See L. 382. 

Line 357. What does “[v]” mean in this context?

The units for 88Sr is voltage (V), which are the measurements of the number of 88Sr atoms hitting the instrument detector (faraday cup). Made change. See L. 384.

Lines 352-366. I found this whole paragraph very difficult to understand, right from the calculation of concentrations through to the categorizations.

Thank you for the feedback. We revised the text to be more clear. See L. 393–409. 

Lines 380-384, Fig. 2 caption. Because broad whitefish data from this study are being compared with data from other species and studies, the geographic scope of the other data needs to be defined in the caption. 

Good suggestion. We update Fig 2 caption. See L. 425–432.

Did all the other data come from the same area of northern Alaska? 

No, some information is from northern Alaska, while other information is from the northern Yukon and Northwest Territories. See L. 428–432.

Also, the broad whitefish point should be a mean +/- SD so that its variability is directly comparable to the other points. Using ranges inflates the relative variability.

Made change. See updated Fig 2.

Lines 390-391. This is poorly worded. Presumably you are talking about relationships between delta13C and delta15N within each of the two tissues? Were the relationships positive or negative?

Lines 390–393 (438–411 in revised manuscript), within the tissue comparison paragraph describes the correlation and linear relationship between tissue types (muscle vs. liver) for δ13Cˈ and δ15N. The relationship between muscle and liver δ13Cˈ and muscle and liver δ15N were both positive. 

Line 401, Fig 3 caption. Suggest indicating that blue is less than or equal to 40 cm and red is greater than or equal to 65 cm. Also, should read “For both isotope ratios, positive values indicate…”

Made change. See L. 449–450.

Lines 431-435. Table 1 provides the same information as Figure 6. I suggest deleting the table.

We deleted the table from the manuscript but included it as a table within the supplementary information section.

Lines 460-463. Table 2 provides the same information as Figure 7. I suggest deleting the table.

We deleted the table from the manuscript but included it as a table within the supplementary information section.

Somewhere in Discussion. Should mention how trophic diversity is measured and how this can influence the interpretation of results. Trophic diversity has both within-individual and among-individual components. In this study, diversity is measured among-individuals but not within-individuals. The among-individual diversity is sometimes interpreted as a measure of individual specialization.

Thank you for the suggestion. We added clarifying text to the discussion section. See L. 516–517, L. 548–552.

Somewhere in Discussion. It should be stated more clearly that delta13C variation can be interpreted in various ways. Delta13C can reflect the pelagic-benthic primary production continuum in larger lakes, and can also reflect the terrestrial-aquatic (or allochthonous-authochthonous) continuum in rivers, or more riverine lakes. Most terrestrial primary production is very 13C depleted, similar to pelagic production. It can also reflect the freshwater-marine continuum, as in the Colville system.

Thank you for the clarifying suggestion. We added suggested text to the manuscript. See L. 628–631. 

Lines 646-670, Acknowledgements. Condense considerably.

Made suggested changes. Please see the condensed acknowledgment section. See L. 707–731.

6. PLOS authors have the option to publish the peer review history of their article (what does this mean?). If published, this will include your full peer review and any attached files.

Do you want your identity to be public for this peer review? For information about this choice, including consent withdrawal, please see our Privacy Policy.

Reviewer #1: No

Reviewer #2: No

---

## [Editor Report · Decision Letter 1]

13 Jun 2022

Broad Whitefish (*Coregonus nasus *) isotopic niches: stable isotopes reveal diverse foraging strategies and habitat use in Arctic Alaska

PONE-D-21-39878R1

Dear Dr. Leppi,

We’re pleased to inform you that your manuscript has been judged scientifically suitable for publication and will be formally accepted for publication once it meets all outstanding technical requirements.

Kind regards,

Giorgio Mancinelli, Ph.D.

Academic Editor

PLOS ONE

Additional Editor Comments (optional):

I have considered the changes made by the authors to the original version of the manuscript, and my conclusions are that they have considerably increased the quality of the ms, making it acceptable for publication in PLOS ONE.
---

## [Editor Report · Acceptance letter]

1 Jul 2022

PONE-D-21-39878R1 

Broad Whitefish (*Coregonus nasus*) isotopic niches: stable isotopes reveal diverse foraging strategies and habitat use in Arctic Alaska 

Dear Dr. Leppi:

I'm pleased to inform you that your manuscript has been deemed suitable for publication in PLOS ONE. Congratulations! Your manuscript is now with our production department. 

Kind regards, 

on behalf of

Dr. Giorgio Mancinelli 

Academic Editor

PLOS ONE